# RWKV: Reinventing RNNs for the Transformer Era

**Bo Peng**[1,2]* **Eric Alcaide**[2,3,4]* **Quentin Anthony**[2,5]*

**Alon Albalak**[2,6] **Samuel Arcadinho**[2,7] **Stella Biderman**[2,8] **Huanqi Cao**[9] **Xin Cheng**[10]
**Michael Chung**[11] **Xingjian Du**[1] **Matteo Grella**[12] **Kranthi Kiran GV**[2,13] **Xuzheng He**[2]
**Haowen Hou**[14] **Jiaju Lin**[1] **Przemysław Kazienko**[15] **Jan Kocoń**[15] **Jiaming Kong**[16]
**Bartłomiej Koptyra**[15] **Hayden Lau**[2] **Krishna Sri Ipsit Mantri**[17] **Ferdinand Mom**[18,19]
**Atsushi Saito**[2,20] **Guangyu Song**[21] **Xiangru Tang**[22] **Bolun Wang**[23] **Johan S. Wind**[24]
**Stanisław Woźniak**[15] **Ruichong Zhang**[9] **Zhenyuan Zhang**[2] **Qihang Zhao**[25,26]
**Peng Zhou**[23] **Qinghua Zhou**[5] **Jian Zhu**[27] **Rui-Jie Zhu**[28,29]

[1]Generative AI Commons [2]EleutherAI [3]U. of Barcelona [4]Charm Therapeutics [5]Ohio State U. [6]U. of C., Santa Barbara

[7]Zendesk [8]Booz Allen Hamilton [9]Tsinghua University [10]Peking University [11]Storyteller.io [12]Crisis24 [13]New York U.

[14]National U. of Singapore [15]Wroclaw U. of Science and Technology [16]Databaker Technology [17]Purdue U. [18]Criteo AI Lab

[19]Epita [20]Nextremer [21]Moves [22]Yale U. [23]RuoxinTech [24]U. of Oslo [25]U. of Science and Technology of China

[26]Kuaishou Technology [27]U. of British Columbia [28]U. of C., Santa Cruz [29]U. of Electronic Science and Technology of China

## Abstract

Transformers have revolutionized almost all natural language processing (NLP) tasks but suffer from memory and computational complexity that scales quadratically with sequence length. In contrast, recurrent neural networks (RNNs) exhibit linear scaling in memory and computational requirements but struggle to match the same performance as Transformers due to limitations in parallelization and scalability. We propose a novel model architecture, Receptance Weighted Key Value (RWKV), that combines the efficient parallelizable training of transformers with the efficient inference of RNNs.

Our approach leverages a linear attention mechanism and allows us to formulate the model as either a Transformer or an RNN, thus parallelizing computations during training and maintains constant computational and memory complexity during inference. We scale our models as large as 14 billion parameters, by far the largest dense RNN ever trained, and find RWKV performs on par with similarly sized Transformers, suggesting future work can leverage this architecture to create more efficient models. This work presents a significant step towards reconciling trade-offs between computational efficiency and model performance in sequence processing tasks. [1]

## 1 Introduction

Deep learning has greatly advanced artificial intelligence, impacting a range of scientific and industrial uses. These often involve complex sequential data

processing tasks such as natural language understanding, conversational AI, time-series analysis, and indirectly sequential formats like images and graphs (Brown et al., 2020; Ismail Fawaz et al., 2019; Wu et al., 2020; Albalak et al., 2022). Predominant among these techniques include RNNs and Transformers (Vaswani et al., 2017), each with specific benefits and drawbacks. RNNs require less memory, particularly for handling long sequences. However, they suffer from the vanishing gradient problem and non-parallelizability in the time dimension during training, limiting their scalability (Hochreiter, 1998; Le and Zuidema, 2016).

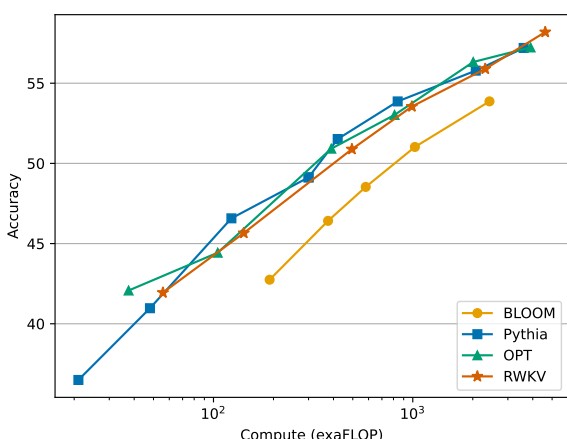

Figure 1: Average performance of RWKV models compared to transformers across twelve NLP tasks. For further details, see section 5.

Transformers emerged as a powerful alternative, adept at managing local and long-range dependencies and supporting parallelized training (Tay et al., 2022). Models such as GPT-3 (Brown et al., 2020), ChatGPT (OpenAI, 2022; Kocoń et al., 2023),

---

* Equal first authorship. Others listed alphabetically.
[1]Code at: https://github.com/BlinkDL/RWKV-LM

| Model | Time | Space |
|-------|------|-------|
| Transformer | $O(T^2d)$ | $O(T^2 + Td)$ |
| Reformer | $O(T \log Td)$ | $O(T \log T + Td)$ |
| Performer | $O(Td^2 \log d)$ | $O(Td \log d + d^2 \log d)$ |
| Linear Transformers | $O(Td^2)$ | $O(Td + d^2)$ |
| AFT-full | $O(T^2d)$ | $O(Td)$ |
| AFT-local | $O(Tsd)$ | $O(Td)$ |
| MEGA | $O(cTd)$ | $O(cd)$ |
| RWKV (ours) | $O(\mathbf{Td})$ | $O(\mathbf{d})$ |

Table 1: Inference complexity comparison with different Transformers. Here $T$ denotes the sequence length, $d$ the feature dimension, $c$ is MEGA's chunk size of quadratic attention, and $s$ is the size of a local window for AFT.

LLaMA (Touvron et al., 2023), and Chinchilla (Hoffmann et al., 2022) showcase the potential of Transformers in NLP. However, the self-attention mechanism's quadratic complexity makes it computationally and memory intensive for tasks involving long sequences and constrained resources. This has stimulated research to enhance Transformers' scalability, sometimes sacrificing some of their effectiveness (Wang et al., 2020; Zaheer et al., 2020; Dao et al., 2022a).

To tackle these challenges, we introduce the Receptance Weighted Key Value (**RWKV**) model, combining the strengths of RNNs and Transformers while circumventing key drawbacks. RWKV alleviates memory bottleneck and quadratic scaling associated with Transformers (Katharopoulos et al., 2020) with efficient linear scaling, while maintaining the expressive properties of the Transformer, such as parallelized training and robust scalability. RWKV reformulates the attention mechanism with a variant of linear attention, replacing traditional dot-product token interaction with more effective channel-directed attention. This implementation, *without approximation*, offers the lowest computational and memory complexity; see Table 1.

The motivation behind RWKV is to balance computational efficiency with expressive capacity in neural networks. It offers a solution for handling large-scale models with billions of parameters, exhibiting competitive performance at a reduced computational cost. Experiments suggest RWKV addresses scaling and deployment challenges in AI, especially for sequential data processing, pointing towards more sustainable and efficient AI models.

Our contributions in this paper are as follows:

- The introduction of RWKV, a novel architec-

ture combining RNNs and Transformer advantages while mitigating their limitations.
- Detailed experiments demonstrating RWKV's performance and efficiency on benchmark datasets for large-scale models.
- The release of pretrained models, from 169 million to 14 billion parameters, trained on the Pile (Gao et al., 2020; Biderman et al., 2022).[2]

## 2 Background

Here we briefly review the fundamentals of RNNs and Transformers.

### 2.1 Recurrent Neural Networks (RNNs)

Popular RNN architectures such as LSTM (Hochreiter and Schmidhuber, 1997) and GRU (Chung et al., 2014) are characterized by the following formulation (shown for LSTM, others can be reasoned similarly):

$$f_t = \sigma_g(W_f x_t + U_f h_{t-1} + b_f), \qquad (1)$$
$$i_t = \sigma_g(W_i x_t + U_i h_{t-1} + b_i), \qquad (2)$$
$$o_t = \sigma_g(W_o x_t + U_o h_{t-1} + b_o), \qquad (3)$$
$$\tilde{c}_t = \sigma_c(W_c x_t + U_c h_{t-1} + b_c), \qquad (4)$$
$$c_t = f_t \odot c_{t-1} + i_t \odot \tilde{c}_t, \qquad (5)$$
$$h_t = o_t \odot \sigma_h(c_t). \qquad (6)$$

Although RNNs can be factored into two linear blocks ($W$ and $U$) and an RNN-specific block (1)–(6), as noted by Bradbury et al. (2017), the data dependency relying on previous time steps prohibits parallelizing these typical RNNs.

### 2.2 Transformers and AFT

Introduced by Vaswani et al. (2017), Transformers are a class of neural networks that have become the dominant architecture for several NLP tasks. Instead of operating on sequences step-by-step like RNNs, Transformers rely on attention mechanisms to capture relationships between all input and all output tokens:

$$\mathrm{Attn}(Q, K, V) = \mathrm{softmax}(QK^\top)V, \qquad (7)$$

where the multi-headness and scaling factor $\frac{1}{\sqrt{d_k}}$ is omitted for convenience. The core $QK^\top$ multiplication is an ensemble of pairwise attention scores

---

[2]https://huggingface.co/RWKV

between each token in a sequence, which can be decomposed as vector operations:

$$\text{Attn}(Q, K, V)_t = \frac{\sum_{i=1}^{T} e^{q_t^\top k_i} \odot v_i}{\sum_{i=1}^{T} e^{q_t^\top k_i}}. \quad (8)$$

AFT (Zhai et al., 2021), alternately formulates

$$\text{Attn}^+(W, K, V)_t = \frac{\sum_{i=1}^{t} e^{w_{t,i} + k_i} \odot v_i}{\sum_{i=1}^{t} e^{w_{t,i} + k_i}}, \quad (9)$$

where $\{w_{t,i}\} \in R^{T \times T}$ is the learned pair-wise position biases, and each $w_{t,i}$ is a scalar.

Inspired by AFT, RWKV takes a similar approach. However, for simplicity, it modifies the interaction weights so that it can be transformed into an RNN. Each $w_{t,i}$ in RWKV is a channel-wise time decay vector multiplied by the relative position and traced backward from current time as it decays:

$$w_{t,i} = -(t - i)w, \quad (10)$$

where $w \in (R_{\geq 0})^d$, with $d$ the number of channels. We require $w$ to be non-negative to ensure that $e^{w_{t,i}} \leq 1$ and the per-channel weights decay backwards in time.

## 3 RWKV

The RWKV model architecture is defined by four fundamental elements that are intrinsic to the time-mixing and channel-mixing blocks:

- $R$: The **Receptance** vector acts as the receiver of past information.
- $W$: The **Weight** signifies the positional weight decay vector, a trainable parameter within the model.
- $K$: The **Key** vector performs a role analogous to $K$ in traditional attention mechanisms.
- $V$: The **Value** vector functions similarly to $V$ in conventional attention processes.

These core elements interact multiplicatively at each timestep, as depicted in Figure 2.

### 3.1 Architecture

The RWKV model is composed of stacked residual blocks. Each block consists of a time-mixing and a channel-mixing sub-block, embodying recurrent structures to leverage past information.

This model uses a unique attention-like score update process, which includes a time-dependent

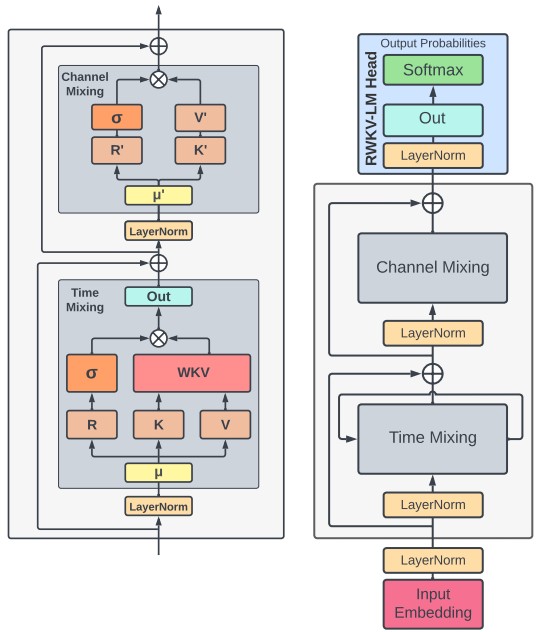

Figure 2: Elements within an RWKV block (left) and the complete RWKV residual block, equipped with a final head for language modeling (right).

softmax operation improving numerical stability and mitigating vanishing gradients (for rigorous proof, see Appendix H). It ensures that the gradient is propagated along the most relevant path. Additionally, layer normalization (Ba et al., 2016) incorporated within the architecture aids in stabilizing the gradients, effectively addressing both vanishing and exploding gradient issues. These design elements not only enhance the training dynamics of deep neural networks but also facilitate the stacking of multiple layers, leading to superior performance over conventional RNN models by capturing complex patterns across different levels of abstraction (see also Appendix I).

#### 3.1.1 Token Shift

In this architecture, all linear projection vectors ($R$, $K$, $V$ in time-mixing, and $R'$, $K'$ in channel-mixing) involved in computations are produced by linear interpolation between current and previous timestep inputs, facilitating a token shift.

The vectors for time-mixing computation are linear projections of linear combinations of the current and previous inputs of the block:

$$r_t = W_r \cdot (\mu_r \odot x_t + (1 - \mu_r) \odot x_{t-1}), \quad (11)$$
$$k_t = W_k \cdot (\mu_k \odot x_t + (1 - \mu_k) \odot x_{t-1}), \quad (12)$$
$$v_t = W_v \cdot (\mu_v \odot x_t + (1 - \mu_v) \odot x_{t-1}), \quad (13)$$

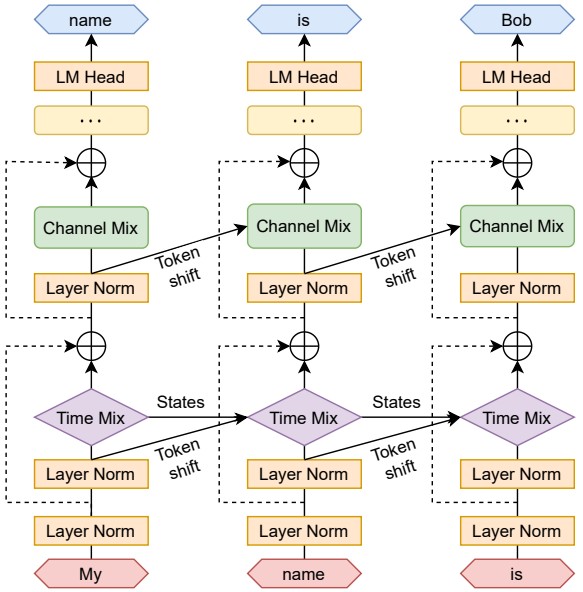

Figure 3: RWKV architecture for language modeling.

as are the channel-mixing inputs:

$$r'_t = W'_r \cdot (\mu'_r \odot x_t + (1 - \mu'_r) \odot x_{t-1}), \quad (14)$$

$$k'_t = W'_k \cdot (\mu'_k \odot x_t + (1 - \mu'_k) \odot x_{t-1}). \quad (15)$$

The token shift is implemented as a simple offset in the temporal dimension at each block using the PyTorch (Paszke et al., 2019) library as `nn.ZeroPad2d((0,0,1,-1))`.

### 3.1.2 WKV Operator

The computation of the $WKV$ operator in our model parallels the method used in Attention Free Transformer (AFT) (Zhai et al., 2021). However, unlike AFT where $W$ is a pairwise matrix, our model treats $W$ as a channel-wise vector that is modified by relative position. In our model, this recurrent behavior is defined by the time-dependent update of the $WKV$ vectors, formalized in the following equation:

$$wkv_t = \frac{\sum_{i=1}^{t-1} e^{-(t-1-i)w+k_i} \odot v_i + e^{u+k_t} \odot v_t}{\sum_{i=1}^{t-1} e^{-(t-1-i)w+k_i} + e^{u+k_t}}. \quad (16)$$

To circumvent any potential degradation of $W$, we introduce a vector $U$ that separately attends to the current token. More information about this can be found in Appendix I.

### 3.1.3 Output Gating

Output gating is implemented in both time-mixing and channel-mixing blocks using the sigmoid of

the receptance, $\sigma(r)$. The output vector $o_t$ post the $WKV$ operator is given by:

$$o_t = W_o \cdot (\sigma(r_t) \odot wkv_t). \quad (17)$$

In the channel-mixing block, a similar operation is performed:

$$o'_t = \sigma(r'_t) \odot (W'_v \cdot \max(k'_t, 0)^2), \quad (18)$$

where we adopt the squared ReLU activation function (So et al., 2021).

### 3.2 Transformer-like Training

RWKV can be efficiently parallelized using a technique called *time-parallel mode*, reminiscent of Transformers. The time complexity of processing a batch of sequences in a single layer is $O(BTd^2)$, primarily consisting of matrix multiplications $W_\lambda$, where $\lambda \in \{r, k, v, o\}$ (assuming $B$ sequences, $T$ maximum tokens, and $d$ channels). In contrast, updating attention scores $wkv_t$ involves a serial scan (see Appendix D for more detail) and has complexity $O(BTd)$.

The matrix multiplications can be parallelized similarly to $W_\lambda$, where $\lambda \in \{Q, K, V, O\}$ in conventional Transformers. The element-wise $WKV$ computation is time-dependent but can be readily parallelized along the other two dimensions (Lei et al., 2018)[3].

### 3.3 RNN-like Inference

Recurrent networks commonly utilize the output at state $t$ as input at state $t + 1$. This usage is also observed in the autoregressive decoding inference of language models, where each token must be computed before being passed to the next step. RWKV takes advantage of this RNN-like structure, known as *time-sequential mode*. In this context, RWKV can be conveniently formulated recursively for decoding during inference, as demonstrated in Appendix D.

### 3.4 Additional Optimizations

**Custom Kernels** To address inefficiencies in the $WKV$ computation arising from the sequential nature of the task when using standard deep learning frameworks, we have developed a custom CUDA

---

[3]For extremely long sequences, more sophisticated methods such as Martin and Cundy (2017) that parallelize over sequence length could be used.

kernel. This kernel enables the execution of a single compute kernel on training accelerators, while all other parts of the model, such as matrix multiplications and point-wise operations, are already inherently parallelizable and efficient.

**Small Init Embedding**   During the initial stage of training a transformer model (Vaswani et al., 2017), we observe that the embedding matrix undergoes slow changes, presenting a challenge for the model to move away from its initial noisy embedding state. To address this issue, we propose an approach that involves initializing the embedding matrix with small values and subsequently applying an additional LayerNorm operation. This accelerates and stabilizes the training process, allowing for the training of deep architectures with post-LN components. The effectiveness of this approach is demonstrated in Figure 9, illustrating improved convergence by enabling the model to quickly transition away from the initially small embedding. This is achieved through small changes occurring in a single step, which subsequently lead to substantial alterations in directions and further notable changes after the LayerNorm operation.

**Custom Initialization**   Building on principles from previous works (He et al., 2016; Jumper et al., 2021), we adopt an initialization strategy where parameters are set to values resembling an identity mapping while breaking symmetry to establish a clear information flow. The majority of weights are initialized to zero, and linear layers do not employ biases. Detailed formulas are given in Appendix E. We observe that the choice of initialization plays a crucial role in both the speed and quality of convergence (refer to Appendix F for further details).

## 3.5   Implementation

RWKV is implemented using the PyTorch Deep Learning Library (Paszke et al., 2019). We integrate additional optimization strategies inspired by DeepSpeed (Rasley et al., 2020) into the system, improving its efficiency and scalability.

The model begins with an embedding layer, as detailed in Section 3.4. Following this are several identical residual blocks arranged sequentially. These are depicted in Figures 2 and 3 and adheres to the principles outlined in Section 3.1.1. After the last block, a simple output projection head, consisting of a LayerNorm (Ba et al., 2016) and a linear projection, is employed for logits generation

for next-token prediction and computation of the cross-entropy loss during training.

## 4   Trained Models and Computing Costs

To demonstrate the scalability of RWKV, we train six models ranging from 169 million to 14 billion parameters as shown in Table 2. All models are trained for one epoch (330 billion tokens) on the Pile (Gao et al., 2020; Biderman et al., 2022).

| Name | Layers | Model Dimension | Parameters | FLOP per token |
|------|--------|-----------------|------------|----------------|
| 169 M | 12 | 768 | $1.693 \times 10^8$ | $2.613 \times 10^8$ |
| 430 M | 24 | 1024 | $4.304 \times 10^8$ | $7.573 \times 10^8$ |
| 1.5 B | 24 | 2048 | $1.515 \times 10^9$ | $2.823 \times 10^9$ |
| 3 B | 32 | 2560 | $2.985 \times 10^9$ | $5.710 \times 10^9$ |
| 7 B | 32 | 4096 | $7.393 \times 10^9$ | $1.437 \times 10^{10}$ |
| 14 B | 40 | 5120 | $1.415 \times 10^{10}$ | $2.778 \times 10^{10}$ |

Table 2: RWKV model architectures and FLOP counts. Further details of these hyperparameters are elaborated upon in Appendix G.

The number of parameters for each model is computed using the formula: # parameters $= 2VD + 13D^2L + D(11L + 4)$ where $V = 50277$ is the vocabulary size, $D$ represents the Model Dimension and $L$ corresponds to the number of layers. FLOPs is for a forward pass for one token. It was calculated as $2(2VD + 13D^2L)$, which is the twice (add and multiply) the number of parameters in linear layers. The backwards pass FLOPs can be approximated as twice that of the forward pass, giving a total of $6(2VD + 13D^2L)$ FLOP per token. Notably, this matches the standard formula for FLOP calculations in transformers Kaplan et al. (2020): FLOP $= 6 \cdot$ [# tokens] $\cdot$ [# parameters].

### 4.1   Additional Training Details

For training, we use the standard Adam optimizer without weight decay, use bfloat16 precision, and train with a context length of 1024 tokens. Further details on hyperparameters are in Appendix G. Diverting from standard practice for transformers, we apply exponential decay to our learning rate. We also incorporate the auxiliary loss introduced by PaLM (Chowdhery et al., 2022), supplementing the standard cross-entropy loss function. This auxiliary loss encourages the softmax normalizer to approximate zero closely. As for the learning rate schedule, it remains constant for the initial iterations, and subsequently decays exponentially.

## 4.2 Scaling Laws

Scaling laws (Kaplan et al., 2020; Henighan et al., 2020; Hoffmann et al., 2022; Muennighoff et al., 2023) in language models refer to the mathematical relationships that describe how the performance of a language model changes with respect to various factors. These factors can include the model size ($N$), dataset size ($D$), or the optimally allocated compute budget ($C_{\min}$). Scaling laws are important for two primary reasons: they allow us to make predictions and plans regarding the costs and performance of large models before they are trained via interpolation and extrapolation (Black et al., 2022; Le Scao et al., 2022) and the contexts in which they fail provides rich feedback on important areas for future research (Wei et al., 2022a; Biderman et al., 2023a).

Previous work on scaling laws for RNNs has claimed that LSTMs do not strictly follow the same log-log linear scaling that transformers do (Kaplan et al., 2020). We train 45 RWKV models for a variety of pairs (dataset, parameters) and find that RWKV *does* follow the same general form of the scaling law that is well established for transformers. Figure 4 shows our results for loss as a function of compute, with the linear fit to the Pareto optimal points holding an $r^2$ value of $0.994$. Even when we extrapolate our curve an additional order of magnitude (blue), we find an extremely good fit with an $r^2$ of $0.875$.

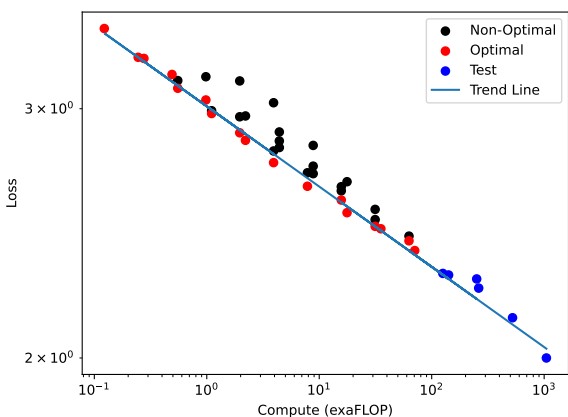

Figure 4: Scaling laws curves for RWKV models

## 5 Evaluations

Having demonstrated the scalability of RWKV models in the previous section, we now turn our attention to their competitiveness with traditional transformers. We focus on two questions:

**Competitiveness** Is RWKV competitive against quadratic transformer architectures with the same amount of compute?

**Long Context** Does increasing the context length of RWKV yield better language modeling loss when RWKV models are trained for context lengths that most open-sourced quadratic transformers *cannot* efficiently process?

### 5.1 NLP Evaluations

To demonstrate that RWKV is competitive with traditional transformers at NLP tasks, we compare with similarly sized models trained for a similar number of tokens (Pythia (Biderman et al., 2023b), OPT (Zhang et al., 2022) and BLOOM (Scao et al., 2022)). All RWKV models were trained for one epoch on the Pile (330B tokens), which is close but not identical to the amount of tokens the Pythia, OPT, and BLOOM models were trained for. Consequently, we compare our models on a *FLOP-matched basis*. We avoid comparing with model trained in the Chinchilla-optimal regime (Hoffmann et al., 2022) or the overtrained regime (Touvron et al., 2023) to ensure the most equitable comparison.

We report results on ARC (both Easy and Challenge) (Clark et al., 2018), BoolQ (Clark et al., 2019), COPA (Roemmele et al., 2018), HeadQA (Vilares and Gómez-Rodríguez, 2019), HellaSwag (Zellers et al., 2019), LAMBADA (Paperno et al., 2016), OpenBookQA (Mihaylov et al., 2018), PIQA (Bisk et al., 2020), ReCoRD (Zhang et al., 2018), SciQ (Johannes Welbl Nelson F. Liu, 2017), and Winogrande (Zellers et al., 2020). Figure 1 shows the average results across all benchmarks. Some individual benchmarks are shown in Fig 5, with the rest in Appendix J.

Additionally, we carried out comparative studies on RWKV and ChatGPT / GPT-4, see Appendix L. They revealed that RWKV is very sensitive to prompt engineering. When the prompts were adjusted (re-ordered) from the ones used for GPT to more suitable for RWKV, the performance (F1) increased even from 44.2% to 74.8%. For sarcasm detection, RWKV outperformed ChatGPT, but was still slightly worse than the SOTA solution.

### 5.2 Extended Context Finetuning

Unlike transformers, RNNs do not have a predefined sequences length when they are created. However in order to efficient make use of compute

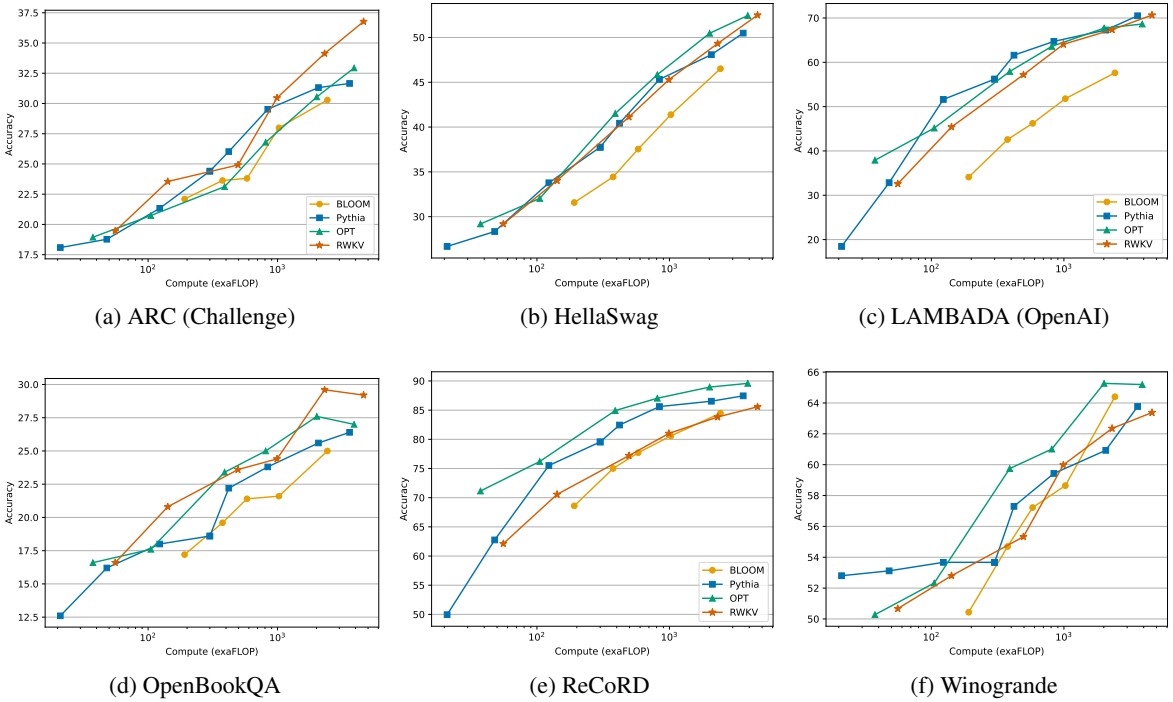

(a) ARC (Challenge)

(b) HellaSwag

(c) LAMBADA (OpenAI)

(d) OpenBookQA

(e) ReCoRD

(f) Winogrande

Figure 5: Zero-Shot Performance of RWKV on common language modeling evaluation benchmarks. Additional plots can be found in Appendix J.

we nevertheless need to preprocess the training data into contexts of the same length. We find that we are able to teach the model how to efficiently handle substantially larger batch sizes by finetuning with progressively increasing sequence length. Specifically, we first double the sequence length from 1024 to 2048 and finetune for 10B tokens from the original pretraining corpus, then we double again to 4096 for 100B tokens from the same corpus, and finally double to 8192 tokens for another 100B tokens from the same corpus. In Fig. 6 we show that increasing context length leads to lower test loss on the Pile, an indication that RWKV can make effective use of long contextual information.

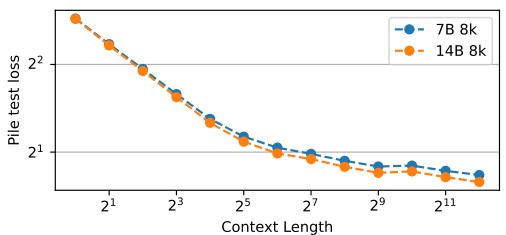

Figure 6: RWKV shows decreasing mean test loss as a function of context length on the Pile (Gao et al., 2020)

## 5.3 Long Context Benchmarks

Additionally, we evaluate our model's ability to handle very long sequences by comparing to state-of-the-art long sequence models on the Long-Range Arena (LRA) benchmark (Tay et al., 2021). LRA is designed to assess the performance of models in handling lengthy context situations. It includes a collection of tasks with sequences ranging from 1,000 to 16,000 tokens, covering various types of data like text, natural language, synthetic images, and mathematical expressions. We apply RWKV on the LRA benchmark and the results are in Appendix J.2. The results show that RWKV performs second only to the S4 model in five datasets.

## 6 Inference Experiments

We benchmark inference requirements according to size and family. Specifically, we evaluate text generation speed and memory requirements on typical compute platforms including CPU (x86) and GPU (NVIDIA A100 80 GB). For all of our inference experiments we use float32 precision and the HuggingFace Transformers (Wolf et al., 2020). We include all model parameters in the parameter count, including both embedding and non-embedding layers. Performance under different quantization setups is left to further work. See Appendix K for

more results.

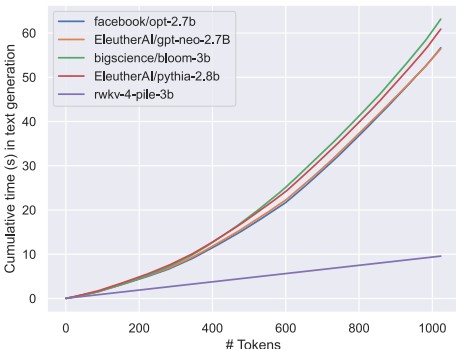

Figure 7: Cumulative time on text generation for LLMs. Unlike transformers, RWKV exhibits linear scaling.

## 7  Future Work

There are several promising directions for future work on the RWKV architecture. Work can be done to increase model expressivity by enhancing the time-decay formulations and exploring initial model states while maintaining efficiency. The RWKV computational efficiency can be further improved by applying a parallel scan in the $wkv_t$ step to reduce the computational cost to $O(B \log(T)d)$. The mechanisms used in RWKV can be applied to encoder-decoder architectures, potentially replacing the cross-attention mechanism. This could be applicable in seq2seq or multimodal settings, thereby enhancing efficiency during both training and inference.

RWKV's state (or *context*) can be leveraged for interpretability, predictability in sequence data, and safety. Manipulating the hidden state could also guide behavior and allow greater customizability through prompt tuning.

## 8  Conclusions

We introduced RWKV, a new approach to RNN models exploiting the potential of time-based mixing components. RWKV introduces several key strategies that allow it to capture locality and long-range dependencies while addressing limitations of current architectures by: (1) replacing the quadratic QK attention with a scalar formulation at linear cost, (2) reformulating recurrence and sequential inductive biases to enable efficient training parallelization and efficient inference, and (3) enhancing training dynamics using custom initializations.

We benchmark the proposed architecture in a wide variety of NLP tasks and show comparable performance to SoTA with reduced cost. Further experiments on expressivity, interpretability, and scaling showcase the model capabilities and draw parallels in behavior between RWKV and other LLMs.

RWKV opens a new route for scalable and efficient architectures to model complex relationships in sequential data. While many alternatives to Transformers have been proposed with similar claims, ours is the first to back up those claims with pretrained models with tens of billions of parameters.

## 9  Limitations

While our proposed RWKV model has demonstrated promising results regarding training and memory efficiency during inference, some limitations should be acknowledged and addressed in future work.

First, the linear attention of RWKV leads to significant efficiency gains but still, it may also limit the model's performance on tasks that require recalling minutiae information over very long contexts. This is due to the funneling of information through a single vector representation over many time steps, compared with the full information maintained by the quadratic attention of standard Transformers. In other words, the model's recurrent architecture inherently limits its ability to "look back" at previous tokens, as opposed to traditional self-attention mechanisms. While learned time decay helps prevent the loss of information, it is mechanistically limited compared to full self-attention.

Another limitation of this work is the increased importance of prompt engineering in comparison to standard Transformer models. The linear attention mechanism used in RWKV limits the information from the prompt that will be carried over to the model's continuation. As a result, carefully designed prompts may be even more crucial for the model to perform well on tasks.

The above RWKV property was confirmed by studies on prompt engineering presented in Appendix L. By changing the order of the information pieces, we were even able to almost double the RWKV performance for some tasks.

## 10  Ethics Statement

In this paper, we present a novel architecture for sequential data processing and prove its effectiveness by building a series of LLMs trained on publicly re-

leased pretraining data (Gao et al., 2020; Biderman et al., 2022) and later fine-tuned on publicly available instructions (Taori et al., 2023; Chaudhary, 2023; Cheung, 2023; Anand et al., 2023; Anonymous, 2023; Yang, 2023; Ji et al., 2023a,b).

As a novel architecture for sequential data, RWKV has the potential to improve sequence-based models across different applications ranging from natural language processing to biomedical data processing or climate modelling. Since the training code is released open source, RWKV contributes to the democratization of AI, levels the playing field, and empowers members of the Open Source community to inspect, study, and finetune RWKV in particular tasks. Moreover, it contributes to advancing the understanding of LLMs capabilities and limitations. A significant amount of work has been devoted to increasing the efficiency of RWKV training so as to minimize its cost and promote accessibility.

As LLMs trained on public data, RWKV's lower inference cost compared to Transformer alternatives makes it more suitable for deployment in consumer and edge hardware, which is a step towards the democratization and distribution of LLMs to the general public, creating better privacy and ownership incentives. It also lowers the resource barrier to Chat assistants and text generation for small and/or underrepresented communities. PreTrained model weights for different sizes ranging from 0.1B to 14B parameters trained on multiple languages are released to increase ease of adoption and allow for the study of emergent phenomena.

On the other hand, with lower resource barriers, the spreading of AI-generated text might become more prevalent. Current RWKV LLMs may exhibit and/or reproduce biases and potentially harmful content present in the data used for training. Nonetheless, mitigation and finetuning strategies discussed for other, large Transformer models should be applicable to RWKV as well.

## Acknowledgements

We thank StabilityAI for the compute used to train our models and for technical support in development of RWKV. We also thank the members of the RWKV and EleutherAI Discord servers for their help and work on further extending the applicability of RWKV to different domains.

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

# A  Author Contributions

All authors contributed to the drafting of this paper. Eric Alcaide and Quentin Anthony organized the paper and its experiments and were involved in all phases of the development process.

**Model Design and Development**    Bo Peng (lead), Matteo Grella, Xuzheng He, Haowen Hou, Jiaming Kong, Johan S. Wind

**Model Training**    Bo Peng

**Scaling Laws Analysis**    Stella Biderman, Bo Peng

**Benchmark Evaluations**    Stella Biderman (lead), Kranthi Kiran GV, Krishna Sri Ipsit Mantri, Atsushi Saito, Qihang Zhao, Peng Zhou, Rui-Jie Zhuåç

**Long Context Experiments**    Xingjian Du, Rui-Jie Zhu, Bolun Wang, Ruichong Zhang, Jian Zhu, Rui-Jie Zhu

**Inference Speed Experiments**    Samuel Arcadinho, Przemysław Kazienko, Qinghua Zhou

**Information Flow Experiments**    Huanqi Cao, Michael Chung, Matteo Grella, Ferdinand Mom, Zhenyuan Zhang

**Chat Experiments**    Jan Kocoń (lead), Przemysław Kazienko, Bartłomiej Koptyra, Hayden Lau, Xiangru Tang, Stanisław Woźniak, Zhenyuan Zhang

**Ethics and Broader Impacts**    Stella Biderman, Guangyu Song

## B  Author Contributions

**Bo Peng**  Original RWKV idea, original code, performance optimizations, original experiments, and trained RWKV models from 0.1B to 14B.

**Eric Alcaide**  Manuscript (initial draft sections 1, C; sections 3, 7 and 8; revision and proofreading; final version ). Figures (2, 3, 3, 8). Experiments section 6. Appendices E, K. Contributions to Appendix M.

**Quentin Anthony**  Manuscript (organization, initial draft sections 1, C, 2; revision and proofreading; final version).

**Alon Albalak**  Manuscript (abstract and sections 1, 9, and 7; proofreading and revision).

**Samuel Arcadinho**  Contributions to Figures 7, 13, and 14. Contributions to Appendix K.

**Stella Biderman**  Performed the scaling laws analysis and evaluated competitor models on benchmark tasks.

**Huanqi Cao**  Manuscript (contributions to 3.2 and 3.3; proofreading and revision). Experiments for Appendix I.

**Xin Cheng**  Manuscript (proofreading and revision). Contributions to Appendix M, J.

**Michael Chung**  Manuscript (contributions to section I; proofreading and revision).

**Xingjian Du**  Evaluation on Long Range Arena Benchmark (TBD until 5.31).

**Matteo Grella**  Manuscript (sections H, I, 8; contributions to sections 1, 7 and 9; proofreading and revision). Contributions to Appendix D.

**Kranthi Kiran GV**  Manuscript (sections C and 5; contributions to section 2; revision and proofreading). Tables K and K. Appendix 4.

**Xuzheng He**  Manuscript (contributions to section 2; proofreading and revision). Contributions to Figure8. Appendix I. Contributions to appendix H.

**Haowen Hou**  Figure 9. Appendix F.

**Jiaju Lin**  RWKV on LRA benchmarking

**Przemysław Kazienko**  Manuscript (proofreading and revision). Contributions to Section 6, 9, and Appendix L.

**Jan Kocon**  Manuscript (Section 1; proofreading and revision). Contributions to Appendix L.

**Jiaming Kong**  Manuscript (revision and proofreading). Appendix H.

**Bartłomiej Koptyra**  Manuscript (revision and proofreading) Contributions to Appendix L.

**Hayden Lau**  Manuscript (contributions to section 1 and 9; proofreading and revision). Contributions to Appendix M.

**Krishna Sri Ipsit Mantri**  Figure 12

**Ferdinand Mom**  Manuscript (contributions to section 1, C, 3.3, I; proofreading and revision). Contributions to Appendix D.

**Atsushi Saito**  Manuscript (sections 2 and 5; contributions to section C). Contributions to Appendix J

**Guangyu Song**  Manuscript (rewrote section 3; final version). Initial draft Ethics Statement).

**Xiangru Tang**  Manuscript (sections C and 2; contributions to abstract; revision and proofreading). Contributions to Appendix M.

**Bolun Wang**    Contributions to Tables 1.

**Johan S. Wind**    RWKV performance optimizations (CUDA), Contributions to Appendix 4.

**Stanisław Woźniak**    Contributions to Appendix L.

**Ruichong Zhang**    Manuscript (proofreading and revision); Contributions to Figure 6 and Appendix M.

**Zhenyuan Zhang**    Manuscript (revision and proofreading). Figure 3. Experiments Appendix I. Contributions to Appendices D and M.

**Qihang Zhao**    Manuscript (proofreading and revision). Contributions to Table 5.

**Peng Zhou**    Contributions to Tables 1 and Table 5.

**Qinghua Zhou**    Manuscript (Proofreading and revision of section 3; Add missing citations in 3.3). Revision of Figures 2 and 12.

**Jian Zhu**    Manuscript (section C; proofreading and revision). Figures 3 and 6.

**Rui-Jie Zhu**    Tables 1 and 5. Experiments for table 5.

## C   Additional Related Work

Recently, a number of techniques have been proposed to address the limitations of transformers.

**Optimizing Attention Mechanism**    Many transformer variants ("x-formers") have been introduced to reduce the complexity of transformers (Tay et al., 2022), including sparse attention (Beltagy et al., 2020; Kitaev et al., 2020; Guo et al., 2022), approximating the full attention matrix (Wang et al., 2020; Ma et al., 2021; Choromanski et al., 2020), combining chunked attention with gating (Ma et al., 2023) and other efficient methods (Katharopoulos et al., 2020; Jaegle et al., 2021).

Some recent works like FlashAttention (Dao et al., 2022a) and others (Rabe and Staats, 2022; Jang et al., 2019) share similarities with RWKV's chunked computation scheme. Despite being memory-efficient, their time complexity remains quadratic or contains chunk size as a hidden factor. In contrast, RWKV achieves better space and time complexity during inference by formulating a linear attention as an RNN.

**Attention Free Models**    Another line of research replaces the attention mechanism with other modules to scale to long sequences. MLP-Mixer and others (Tolstikhin et al., 2021; Liu et al., 2021) propose replacing attention by Multi-Layer Perceptrons (MLPs) in computer vision tasks. The Attention Free Transformer (AFT) (Zhai et al., 2021) and HrrFormer (Alam et al., 2023) replaces dot-product self-attention with a computationally efficient alternative. None of these models have been successfully scaled to the point where drawing comparisons with transformer-based large language models makes sense.

There has also been substantial research into state space models (SSM) (Gu et al., 2021) and its variants (Dao et al., 2022b; Gupta et al., 2022; Poli et al., 2023). In contrast to the preceding models, SSM and its successors have shown substantial progress towards efficient scaling. Simultaneously with this work, Poli et al. (2023) train SSM-based models with 125 million and 355 million parameters and show that the performance is on-par with a transformer that uses a mix of local and global attention (Black et al., 2021).

**Advances in RNNs**    Inspired by the success of transformers, RNN-style (Hochreiter and Schmidhuber, 1997; Chung et al., 2014) recursive components have also been modified to increase context length, such as the Recurrent Memory Transformer (Bulatov et al., 2022, 2023) and Linear Recurrent Units (Orvieto et al., 2023). Most similar to our work, the Quasi-Recurrent neural network (QRNN) (Bradbury et al., 2017) uses both convolutional layers and recurrent pooling functions across timesteps and channels. While QRNN utilizes convolutional filters with fixed sizes, RWKV employs a time-mixing module as an attention mechanism with time-decaying factors. Different from the element-wise pooling in QRNN, RWKV includes a parametrized channel-mixing module that is parallelizable.

## D  Time-Mixing Block as an RNN Cell

As stated in 3.3, the RWKV time-mixing block can be formulated as an RNN, as the $WKV$ computation can be written in such a recursive form:

$$a_0, b_0 = 0, \tag{19}$$

$$wkv_t = \frac{a_{t-1} + e^{u+k_t} \odot v_t}{b_{t-1} + e^{u+k_t}}, \tag{20}$$

$$a_t = e^{-w} \odot a_{t-1} + e^{k_t} \odot v_t, \tag{21}$$

$$b_t = e^{-w} \odot b_{t-1} + e^{k_t}. \tag{22}$$

The dataflow of the RNN-like time-mixing is shown in Fig. 8, where the hidden states $h$ is the numerator-denominator tuple $(a, b)$. To avoid overflow in calculating $e^{k_t}$, a numerical trick is used in the official implementation. Noticing that $a_1 = e^{k_1} \odot v_1$ and $b_1 = e^{k_1}$, we set $a_1' = v_1, b_1' = 1, p_1 = k_1$, where $p_t$ stores the shared exponents of $a_t$ and $b_t$. Now the above recursion can be converted into a numerical safe version, for each time step $t > 1$:

$$q := \max(p_{t-1}, u + k_t), \tag{23}$$

$$wkv_t = \frac{e^{p_{t-1}-q} \odot a_{t-1}' + e^{u+k_t-q} \odot v_t}{e^{p_{t-1}-q} \odot b_{t-1}' + e^{u+k_t-q}}. \tag{24}$$

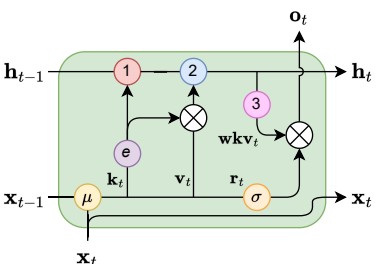

Figure 8: RWKV time-mixing block formulated as an RNN cell. Color codes: yellow ($\mu$) denotes the token shift, red (1) denotes the denominator, blue (2) denotes the numerator, and pink (3) denotes the fraction computations in 16. $h$ denotes the numerator-denominator tuple.

The update to $a_t', b_t'$, and their shared exponent is also carried out in a similar fashion:

$$q' := \max(p_{t-1} - w, k_t), \tag{25}$$

$$a_t' = e^{p_{t-1}-w-q'} \odot a_{t-1}' + e^{k_t-q'} \odot v_t, \tag{26}$$

$$b_t' = e^{p_{t-1}-w-q'} \odot b_{t-1}' + e^{k_t-q'}, \tag{27}$$

$$p_t = q'. \tag{28}$$

The RWKV model has an internal state that stores some previous information. In each layer, the internal state consists five parts, each of which is a vector with $D$ numbers, where $D$ is the model dimension. The five parts are:

- The current input of the Time-mix block $x_t$;
- The current input of the Channel-mix block $y_t$;
- The numerator of the $WKV$ value $a_t'$, as defined in equation (26);
- The denominator of the $WKV$ value $b_t'$, as defined in equation (27);
- An auxiliary state $p_t$ in (28), which is used for $WKV$ computation to maintain numerical precision.

Which yields a total size of $5DL$ parameters. It is worth noting that in an algebraic context with infinite precision, the helper state $p_t$ can be ignored, and the $WKV$ numerator and denominator can be computed directly using equations (21) and (22), reducing the size of the internal state to $4DL$.

## E  Parameter initializations

We describe the specific parameter initializations below and motivate the design choices. Parameters belonging to residual blocks are often adjusted by layer depth and total number of layers. Let # denote the vocabulary size, $s$ denote the embedding dimension, $d$ denote the hidden size (we use $d = 4s$), $L$ the number of layers, $l$ the layer index (from 0 to $L - 1$), we use the following initializations:

- Embeddings are initialized to $\mathcal{U}\,(\pm 1 \times 10^{-4})$ as explained in 3.4
- For the time-mixing blocks (11, 12, 13), initializations are $\mu_{k_i} = \left(\frac{i}{s}\right)^{1-\frac{l}{L}}$, $\mu_{v_i} = \left(\frac{i}{s}\right)^{1-\frac{l}{L}} + \frac{0.3l}{L-1}$ and $\mu_{r_i} = \frac{1}{2} \cdot \left(\frac{i}{s}\right)^{1-\frac{l}{L}}$
- For the channel-mixing blocks (14, 15), $\mu_{k_i}$ and $\mu_{r_i}$ are initialized to $\left(\frac{i}{s}\right)^{1-\frac{l}{L}}$
- $w_i$ (16), also known as "time decay", is initialized to $-5 + 8 \cdot \left(\frac{i}{d-1}\right)^{0.7+\frac{1.3l}{L-1}}$. Intuitively, it is the discount factor applied to previous tokens over time.
- $u_i$ (16), also known as "bonus", is set to $0.5 \cdot (((i+1) \mod 3) - 1) + \log 0.3$. It is the special weighting applied to the current token in equation 16. The alternating zigzag pattern initially creates subtle variations in the tensor elements, which are intended to help the model treat different dimensions of the embedding distinctively.
- $W_o$ (17) (time-mixing) and $W_v$ (channel-mixing) are initialized to $\mathcal{N}(0, \sqrt{\frac{d}{s}} = 2)$
- All other $W_r, W_k, W_v$ weights are initialized to 0 so the model can start learning from the beginning without noisy signals.
- All LayerNorm weights start from 1 and biases from 0.

## F   Small Init Embedding

This section presents the experimental validation of small initialization embedding. The experimental setup is as follows. In the baseline configuration, the parameters are initialized using a normal distribution with a mean of 0.0 and a standard deviation of 0.02, which is a commonly used initialization method in models like BERT and GPT. On the other hand, in the small initialization of the embedding (small init emb) experiment, the parameters are initialized using a uniform distribution with a range of 1e-4, which is slightly different from RWKV where a normal distribution with a standard deviation of 1e-4 is used. However, this difference is negligible and does not affect our conclusions. The experiments were conducted with a batch size of 400. As depicted in Figure 9, the loss curve for the small init emb exhibits a faster rate of decrease and convergence compared to the traditional initialization using a normal distribution.

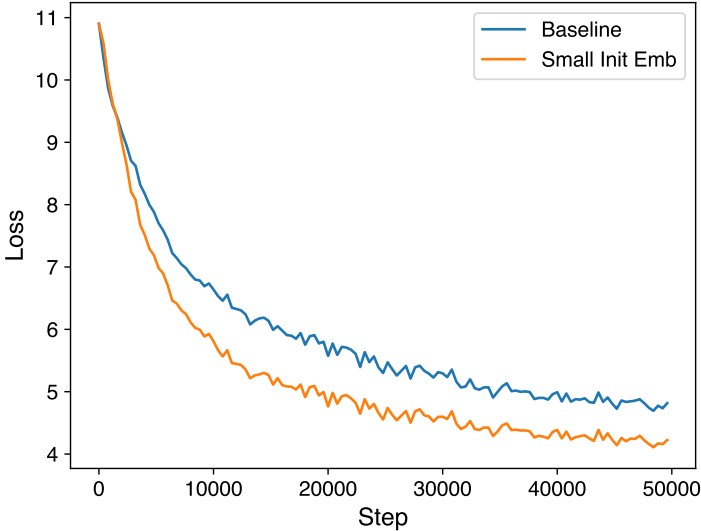

Figure 9: Effect of small initialization embedding.

## G   Hyperparameters

To train the models mentioned, we use $\epsilon = (0.9, 0.99)$ without weight decay for the Adam optimizer, and switch batch size dynamically between 128 or 256 sequences, each of 1024 tokens. We further organize

| Model | 169M | 430M | 1.5B | 3B | 7B | 14B |
|---|---|---|---|---|---|---|
| Init LR | 0.0006 | 0.0004 | 0.0003 | 0.00015 | 0.00015 | 0.0001 |
| Warmup Mini-Epochs | 361 | 411 | 443 | 451 | 465 | 544 |
| End LR | 0.00001 | 0.00001 | 0.00001 | 0.00001 | 0.00001 | 0.000007 |

Table 3: Hyperparameters for our learning rate (LR) schedule of the pretrained models.

the training into multiple mini-epochs, each of 40320 samples, to guide our learning rate schedule. The training process takes 8043 mini-epochs to make one pass over the Pile. The initial warming up mini-epochs have a constant learning rate of "Init LR". After the warming up mini-epochs, the learning rate exponentially decays until in the last mini-epoch, in which the model finishes training on the entire Pile, the learning rate arrives at the "End LR". The related hyperparameters are shown in Table 3.

## H  Gradient Stability in RWKV

In this section, we present a mathematical description of the gradient stability property in RWKV, focusing specifically on the time-mixing block. By gradient stability we mean that if the inputs $x_t$ are bounded and the model parameters are fixed, then the gradients with respect to $W_k$ and $W_v$ are uniformly bounded for all $T$ (thus not exploding). Consequently, we can control the amount each $x_t$ contributes to the gradient at $T$ in a naturally decaying fashion by the weight decay mechanism $w$ (thus not vanishing unless desired).

First, we make the simplification that there are no token shifts, this will not affect the final conclusion. In this scenario, $wkv_T$ can be written as

$$wkv_T = \frac{\sum_{t=1}^{T} K_t^e \odot v_t}{\sum_{t=1}^{T} K_t^e} = \mathrm{E}(v_t) = \frac{\mathrm{S}(v_t)}{\mathrm{S}(1)}, \tag{29}$$

where

$$v_t = W_v x_t, \quad \frac{\partial (v_t)_i}{\partial (W_v)_{i,j}} = (x_t)_j,$$

$$K_t^e = e^{W_k x_t + w_{T,t}}, \quad \frac{\partial (K_t^e)_i}{\partial (W_k)_{i,j}} = (x_t)_j (K_t^e)_i,$$

and $\mathrm{S}(\cdot)$ and $\mathrm{E}(\cdot)$ are shorthand for denoting sums and averages over weights $K_t^e$.

The loss function at position $T$ can be written as

$$L_T = l(f(wkv_T), y_T). \tag{30}$$

Because $wkv_T$ relates to $(W_k)_{i,j}$ and $(W_v)_{i,j}$ only through the $i$-th channel $(wkv_T)_i$, we have

$$\frac{\partial L_T}{\partial (W_v)_{i,j}} = \frac{\partial L_T}{\partial (wkv_T)_i} \frac{\partial (wkv_T)_i}{\partial (W_v)_{i,j}}. \tag{31}$$

The first part of the above equation contains trivial operations like output layers, and other layers of time-mixing, which can be proven inductively. The second part of the above equation can be bounded as

$$\left| \frac{\partial (wkv_T)_i}{\partial (W_v)_{i,j}} \right| = \left| \frac{\partial \mathrm{E}_i[(v_t)_i]}{\partial (W_v)_{i,j}} \right|$$
$$= |\mathrm{E}_i[(x_t)_j]| \leq \max_t |(x_t)_j|, \tag{32}$$

which is irrelevant to $T$. Similarly,

$$
\begin{aligned}
\frac{\partial (wkv_T)_i}{\partial (W_k)_{i,j}} &= \partial \frac{S_i[(v_t)_i]}{S_i(1)} / \partial (W_k)_{i,j} \\
&= \frac{S_i[(x_t)_j (v_t)_i]}{S_i(1)} - \frac{S_i[(x_t)_j] S_i[(v_t)_i]}{S_i(1)^2} \\
&= E_i[(x_t)_j (v_t)_i] - E_i[(x_t)_j] E_i[(v_t)_i] \\
&= \mathrm{cov}_i((x_t)_j, (v_t)_i)
\end{aligned}
\tag{33}
$$

can also be bounded. Note that $wkv$'s softmax operation contains at least two non-zero terms ($u$ and $w$), so the above "covariance" will not degenerate into 0.

## I   Model Behavior Visualization

The right plot illustrates the time decays ($e^{-w}$) in each layer of the RWKV-169M model, sorted along the channel axis. Notably, several decays in the last layers are very close or equal to one, implying that certain information is preserved and propagated throughout the model's temporal context. Meanwhile, many decays in the initial layer are close to zero, which corresponds to local operations in $wkv$ (16), likely to be associated with tasks such as text parsing or lexical analysis. (Note that the local operations in $wkv$ are due to the extra parameter $u$, when $e^{-w}$ is degenerated into 0.) These patterns of time decays are partly learned, but also come from parameter initialization as it speeds up training.

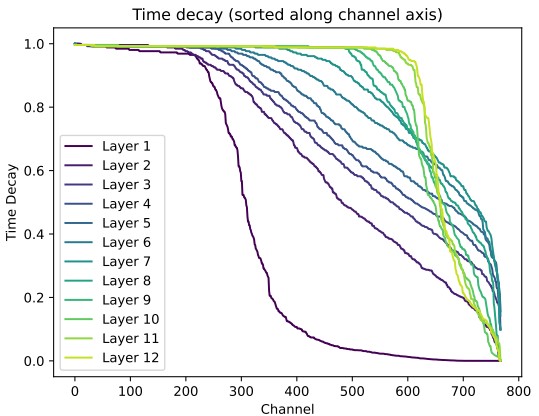

Figure 10: Model behavior visualizations of RWKV.

The plot below shows the information retrieval and propagation path in the RWKV-430M model. The experiment follows the *causal trace* method introduced by Meng et al. (2022), where we

1. Run the model once, and record all states and activation of each layer during the computation;
2. Corrupt the input embeddings of the subject using noise ("The Eiffel Tower" in this example);
3. Restore the states and activation of a certain layer at a certain token during the computation, and record the log-probability of the model outputting the correct answer ("Paris").

Unlike transformers, RWKV relies on the recursive propagation of information in the time dimension. In this case, the fact that the Eiffel Tower is located in Paris is retrieved in layer 4 just after the model sees "The Eiffel". It is then passed down to the subsequent layers. In layer 20, mostly, the information is propagated through time until reaching where it is needed. Finally, at the token "of", it is passed down to the last layer for outputting the answer.

## J   Additional Evaluations

### J.1   Further details on NLP tasks

We evaluate on the following tasks:

**ARC (Clark et al., 2018)**   A dataset designed for multiple-choice question answering, encompassing science exam questions ranging from third grade to ninth grade. It has Easy and Challenge subsets that we report results on separately.

**BoolQ (Clark et al., 2019)**   A binary yes/no question answering benchmark.

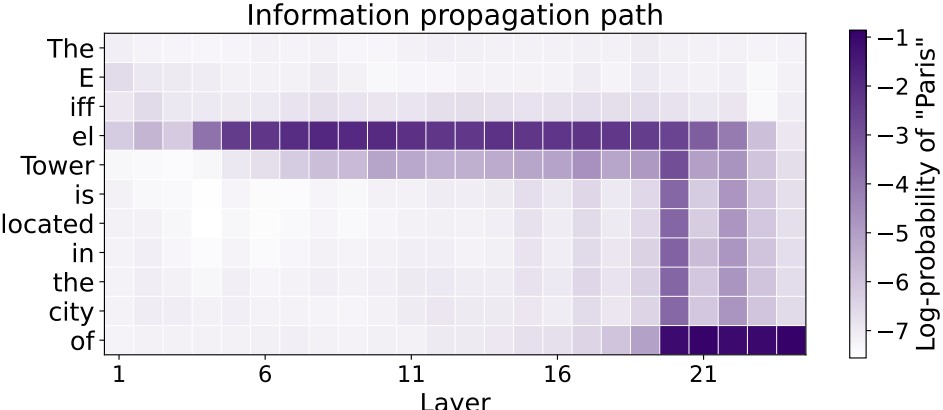

Figure 11: Model behavior visualizations of the RWKV model.

**COPA** **(Roemmele et al., 2018)** A dataset to evaluate achievement in open-domain commonsense causal reasoning.

**HeadQA** **(Vilares and Gómez-Rodríguez, 2019)** A benchmark consisting of graduate-level questions encompassing various fields such as medicine, nursing, biology, chemistry, psychology, and pharmacology.

**HellaSwag** (Zellers et al., 2019) A novel benchmark for commonsense Natural Language Inference (NLI) which is build by adversarial filtering against transformer models.

**LAMBADA (Paperno et al., 2016)** A benchmark dataset that evaluates the model's contextual reasoning and language comprehension abilities by presenting context-target pairs, where the objective is to predict the most probable target token. We follow standard practice and use the untokenized version created by OpenAI (Brown et al., 2020).

**OpenBookQA** **(Mihaylov et al., 2018)** A QA dataset to evaluate human comprehension of a subject by incorporating open book facts, scientific knowledge, and perceptual common sense, drawing inspiration from open book exams.

**PIQA (Bisk et al., 2020)** A benchmark for the task of physical common sense reasoning, which consists of a binary choice task that can be better understood as a set of two pairs, namely (Goal, Solution).

**ReCoRD** **(Zhang et al., 2018)** A benchmark for evaluating commonsense reasoning in reading comprehension by generating queries from CNN/Daily Mail news articles and requiring text span answers from corresponding summarizing passages.

**SciQ** **(Johannes Welbl Nelson F. Liu, 2017)** A multiple-choice QA dataset which was created using an innovative approach to gather well-crafted multiple-choice questions that are focused on a specific domain.

**Winogrande** **(Zellers et al., 2020)** A dataset designed to evaluate the acquisition of common sense reasoning by neural language models, aiming to determine whether we are accurately assessing the true capabilities of machine common sense.

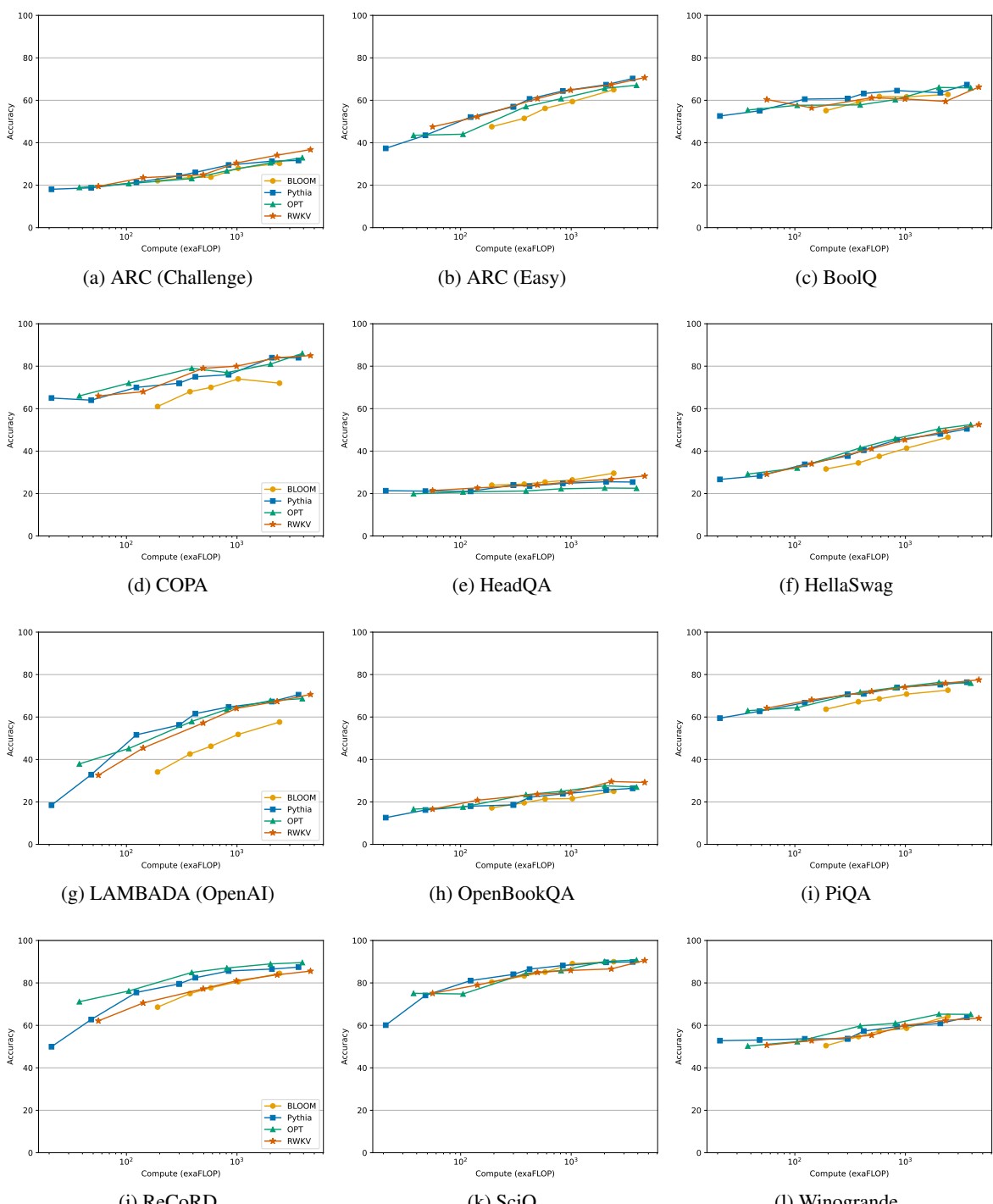

Figure 12: Zero-Shot Performance of RWKV on common language modeling evaluation benchmarks.

## J.2 Evaluation on Long Range Arena

The Long-Range Arena (LRA) benchmark (Tay et al., 2021) is designed to assess the performance of models in handling lengthy context situations. It includes a collection of tasks with sequences ranging from 1,000 to 16,000 tokens, covering various types of data like text, natural language, synthetic images, and mathematical expressions. We apply RWKV on the LRA benchmark and the report results are in Table 4. Other models' performances are directly cited from Gu et al. (2022); Alam et al. (2023).

Table 4: Evaluation on Long Range Arena. Other models reported in the literature (Gu et al., 2022; Alam et al., 2023). **Bolded** values are the best.

| MODEL | LISTOPS | TEXT | RETRIEVAL | IMAGE | PATHFINDER | PATH-X | AVG |
|---|---|---|---|---|---|---|---|
| Transformer | 36.37 | 64.27 | 57.46 | 42.44 | 71.40 | ✗ | 53.66 |
| Reformer | 37.27 | 56.10 | 53.40 | 38.07 | 68.50 | ✗ | 50.56 |
| BigBird | 36.05 | 64.02 | 59.29 | 40.83 | 74.87 | ✗ | 54.17 |
| Linear Trans. | 16.13 | 65.90 | 53.09 | 42.34 | 75.30 | ✗ | 50.46 |
| Performer | 18.01 | 65.40 | 53.82 | 42.77 | 77.05 | ✗ | 51.18 |
| FNet | 35.33 | 65.11 | 59.61 | 38.67 | 77.80 | ✗ | 54.42 |
| Nyströmformer | 37.15 | 65.52 | 79.56 | 41.58 | 70.94 | ✗ | 57.46 |
| Luna-256 | 37.25 | 64.57 | 79.29 | 47.38 | 77.72 | ✗ | 59.37 |
| Hrrformer | 39.98 | 65.38 | 76.15 | 50.45 | 72.17 | ✗ | 60.83 |
| S4 | **59.60** | **86.82** | **90.90** | **88.65** | **94.20** | **96.35** | **86.09** |
| RWKV | 55.88 | 86.04 | 88.34 | 70.53 | 58.42 | ✗ | 72.07 |

The results show that RWKV performs second only to the S4 model in five datasets. While RWKV substantially underpreforms S4 on Image, Pathfinder, and Path-X, on the problems related to natural language and computer code processing RWKV performs on par with S4 or nearly so.

## J.3 Enwik8 Perplexity

We also evaluate our model in terms of perplexity on the Enwik8 dataset. Baseline comparisons are made with Reformer (Kitaev et al., 2020), Synthesizer (Tay et al., 2020) (the best performing dense version), Linear Transformer (Katharopoulos et al., 2020), Performer (Choromanski et al., 2020). $L, d,$ and $T$ denote the number of blocks (network depth), dimension of features, and sequence length, respectively. Both Linear Transformer and Performer are implemented with customized CUDA kernels (github.com/idiap/fast-transformers), and all other models are implemented in native Pytorch. [1] No weight decay nor dropout was used. [2] Trained with AdamW and weight decay set to 0.1, dropout of 0.1, batch size of 16, and initial learning rate of 6e-4.

| Method | L | d | T | Train bpc | Test bpc | Time Complexity | Space Complexity |
|---|---|---|---|---|---|---|---|
| Transformer | 12 | 512 | 1024 | 0.977 | 1.137 | $O(T^2d)$ | $O(T^2 + Td)$ |
| Transformer | 24 | 256 | 1024 | 1.039 | 1.130 | $O(T^2d)$ | $O(T^2 + Td)$ |
| Reformer | 12 | 512 | 1024 | 1.040 | 1.195 | $O(T \log T d)$ | $O(T \log T + Td)$ |
| Synthesizer | 12 | 512 | 1024 | 0.994 | 1.298 | $O(T^2d)$ | $O(T^2 + Td)$ |
| Linear Transformer | 12 | 512 | 1024 | 0.981 | 1.207 | $O(Td^2)$ | $O(Td + d^2)$ |
| Performer | 12 | 512 | 1024 | 1.002 | 1.199 | $O(Td^2 \log d)$ | $O(Td \log d + d^2 \log d)$ |
| AFT-simple | 12 | 512 | 1024 | 1.046 | 1.209 | $O(Td)$ | $O(Td)$ |
| RWKV-RNN[1] | 6 | 512 | 1024 | 0.720 | - | $O(\mathbf{T}\mathbf{d})$ | $O(\mathbf{d})$ |
| RWKV-RNN[2] | 12 | 512 | 1024 | 1.010 | 1.178 | $O(\mathbf{T}\mathbf{d})$ | $O(\mathbf{d})$ |

Table 5: Enwik8 results, measured in bits per character (bpc).

## K Inference results

Figures 13 and 14 illustrate, respectively, the results on time (s) and memory (RAM, VRAM) requirements for LLM inference in *float32* precision. We benchmark the following model families and sizes:

- **RWKV**: 169m, 430m, 1.4b, 3b, 7b, 14b
- **Bloom** (Scao et al., 2022): 560m, 1b, 3b
- **OPT** (Zhang et al., 2022): 125m, 350m, 1.3b, 2.7b, 6.7b, 13b
- **GPT-Neo** (Black et al., 2021): 125m, 1.3b, 2.7b
- **Pythia** (Biderman et al., 2023b): 160m, 410m, 1.4b, 2.8b, 6.7b, 12b

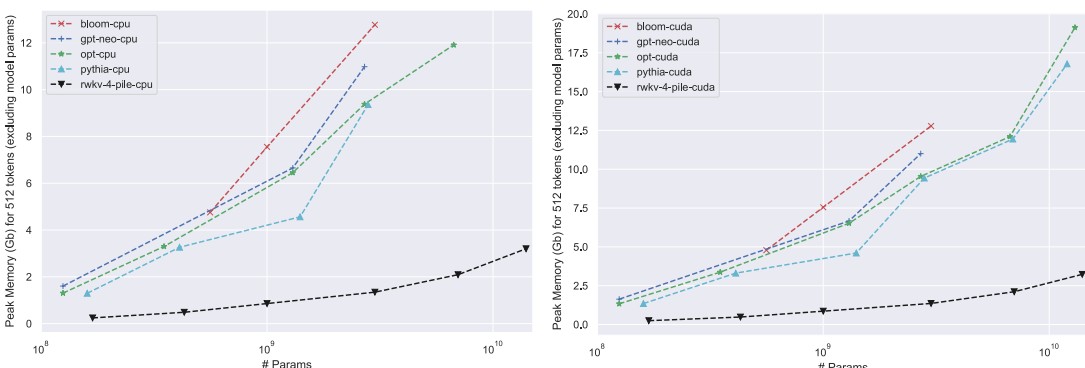

Figure 13: Text generation inference memory (CPU RAM, GPU VRAM) for LLMs. Model parameters are not accounted.

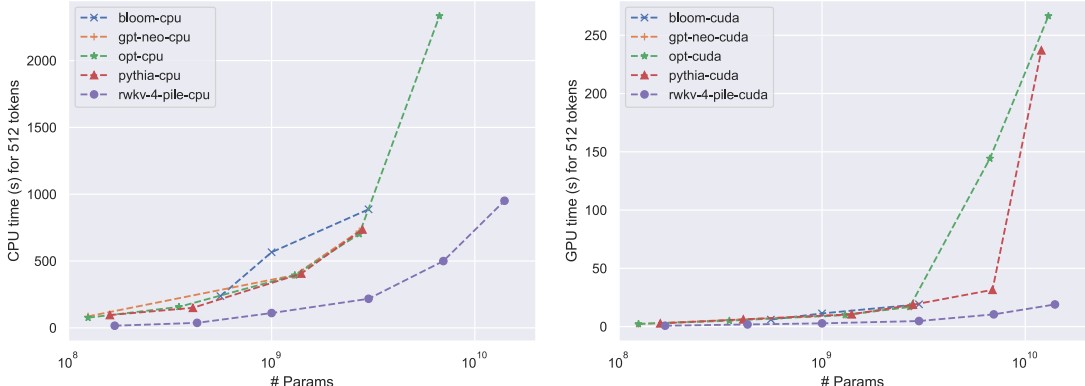

Figure 14: Text generation inference time for LLMs.

| Task Name | Measure | ChatGPT | GPT-4 | RWKV-GPT | RWKV-adapted | SOTA |
|-----------|---------|---------|-------|----------|--------------|------|
| RTE | F1 Macro | 88.1 | **91.3** | 44.2 | 74.8 | 92.1 |
| WNLI | Accuracy | 81.7 | **91.6** | 47.9 | 49.3 | 97.9 |
| GoEmotions | F1 Macro | **25.6** | 23.1 | 7.9 | 7.9 | 52.8 |
| PolEmo2 | F1 Macro | **44.1** | 41.0 | 38.2 | 40.9 | 76.4 |

Table 6: ChatGPT, GPT-4 and RWKV-4-Raven-14B reasoning performance comparison in RTE (Wang et al., 2019), WNLI (Wang et al., 2018), GoEmotions (Demszky et al., 2020), and PolEmo2 (Kocoń et al., 2019) benchmarks. RWKV GPT prompts were primarily used for ChatGPT in (Kocoń et al., 2023). SOTA is provided as a supplementary reference.

| Task Name | Measure | ChatGPT | RWKV-adapted | SOTA |
|-----------|---------|---------|--------------|------|
| Aggression | F1 Macro | **69.10** | 56.66 | 74.45 |
| MathQA | Accuracy | **71.40** | 5.43 | 83.20 |
| Sarcasm | F1 Macro | 49.88 | **50.96** | 53.57 |
| TweetSent | F1 Macro | **63.32** | 52.50 | 72.07 |
| Unhealthy | F1 Macro | **45.21** | 43.30 | 50.96 |

Table 7: ChatGPT and RWKV-4-Raven-14B performance comparison in Aggresion (Wulczyn et al., 2017), Sarcasm (Siddiqui, 2019), Unhealthy (Price et al., 2020), MathQA (Cobbe et al., 2021), and TweetSent (Barbieri et al., 2020) benchmarks. SOTA is provided as a supplementary reference.

## L  Importance of prompt construction and comparison to GPT models

Inspired by Kocoń et al. (2023), we compared the zero-shot performance of the RWKV-4-Raven-14B with ChatGPT (access in February 2023) and GPT-4 using several known NLP tasks, i.e., recognizing textual entailment (RTE), Winograd Natural Language Inference (WNLI), and recognizing emotions elicited in readers (GoEmotions and PolEmo2). Each model got the same prompts manually chosen to receive proper responses from the ChatGPT model. As shown in Tab. 6, RWKV performs significantly worse than ChatGPT and GPT-4 in several specific tasks. We suspect that this disparity is likely caused by the choice of prompts used to generate the answers since the prompts are written in natural language and do not take into account that RWKV, as an RNN, is unable to look back inside an instruction.

When the instruction style was adapted (re-ordered) to respect that RNNs are not capable of "retrospective processing", the quality may significantly change, e.g., for RTE (Wang et al., 2019) F1 Macro increased from 44.2% to 74.8%. We hypothesize that RWKV models are more sensitive to the position of the components in the context, as RNN-based architectures cannot look back and readjust the weight of previous information. For better performance, the desired information should be placed *after* the main question.

> **An example ChatGPT prompt for recognizing textual entailment (RTE)**
>
> Having premise <here is a premise> judge if the following hypothesis <here is a hypothesis> is logically connected with the premise. Answer "entailment" if yes, or "not_entailment" if no.

> **A re-ordered RWKV prompt for RTE taking into account the nature of the RNN**
>
> Can you tell me if the hypothesis is entailment or is not entailment to the premise?
> premise: <here is a premise>
> hypothesis: <here is a hypothesis>

While separating the instruction from the input is relatively easy to do, some other aspects of prompt engineering are harder to quantify. For that purpose, we also tested the approach of stating the input after

the question on multiple other tasks, i.e., aggression and sarcasm detection, classification of unhealthy (offensive) texts, mathematical Q&A, and sentiment analysis, see Tab. 7. The results suggest that better prompts might reduce the disparity between models. Raven achieves comparable results to ChatGPT on unhealthy conversation detection and even surpasses it on the sarcasm detection dataset. While such an approach to prompting looks necessary, it is not enough in itself to replace the capability of having free access to the whole context. Therefore, prompt engineering seems to be significantly more important for the RNN models rather than for standard transformers. It is entirely possible that good prompts to RNN models do not mean additional restrictions, but should simply be constructed using completely different guidelines. The authors of the aforementioned paper (Kocoń et al., 2023)[4] perform chain-of-thought to improve results on the MathQA dataset. Even including this approach, the Raven model achieved a very low accuracy of 5.43%. Without it, the model performed even worse, performing only very basic and simple calculations and achieving 4.13% accuracy. Raven struggled with questions that required intermediate results. It is likely that the order of information presented in the math questions inside the dataset poses a challenge for the RWKV model. It is yet to be seen if prompt engineering can address this issue. This further emphasizes the importance of the order of information the model receives.

---

**Template used to prompt the Raven model in MathQA with chain-of-thought**

Write the reasoning and highlight the answer to the question at the end in the format: "Answer: ".
The question is: <here is a question>

---

**Template used to prompt the Raven model in MathQA without chain-of-thought**

Write the answer to the math question in the format: "Answer: ".
The question is: <here is a question>

---

## M   Cases

In this part, we present a few instances of outputs produced by RWKV model using a Chat interface[56].

---

**RWKV-4-Raven-14B**

**Tell me about ravens.**

Ravens are large, black birds with a distinctive white head and neck. They are found in most parts of the world, including North America, Europe, Asia, and Australia. Ravens are known for their intelligence and problem-solving abilities. They are also considered to be symbols of death and transformation in many cultures. Ravens are often associated with the afterlife or death because they have been known to eat carrion or even other birds. In some cultures, ravens are also believed to be messengers of the gods or guardians of treasure.

---

[4]This is in line with the idea discussed in (Wei et al., 2022b)
[5]ttps://github.com/BlinkDL/ChatRWKV
[6]https://huggingface.co/spaces/BlinkDL/ChatRWKV-gradio

**Explain the concept of machine learning to a 10-year-old.**

Machine learning is like a super-smart computer that can learn on its own. It can recognize patterns and make decisions based on what it has learned. Just like how you might remember something from the past, the computer can remember what it's learned and use it to make predictions or decisions in the future. It's like having a smart friend who always knows what to do!

**Write a short poem describing nature's beauty in springtime.**

Nature's beauty in springtime is rare,
A symphony of colors, a gentle surprise.
The flowers bloom, the birds sing and sing,
As the sun shines down, warming our skin.
The trees sway in the gentle breeze,
As we take in their beauty with ease.
The sky is blue, the clouds are white,
A sight to behold, it's truly divine.

**Explain the concept of machine learning to a 10-year-old.**

**Below is the numerical stable formula for updating the state variable in RWKV:**

$$a_t^* = e^{p-q}a_{t-1} + e^{u+k_t-q}v_t$$

$$b_t^* = e^{p-q}b_{t-1} + e^{u+k_t-q}$$

$$wkv_t = \frac{a_t^*}{b_t^*}$$

**Please write out the corresponding pytorch code to do so, where both $a_t$ and $b_t$ are torch.Tensor([n])**

Here is the PyTorch code to update the state variable in RWKV using the given formula:

```python
import torch
def update_state(a, b, u, v, p, q):
    a_t = torch.exp(p - q) * a + torch.exp(u + k - q) * v
    b_t = torch.exp(p - q) * b + torch.exp(u + k - q)
    return a_t, b_t
a, b = torch.randn(2, 3)
u = torch.randn(2)
v = torch.randn(2)
wkv = torch.randn(2)
a_t, b_t = update_state(a, b, u, v, wkv, torch.randn(2))
print("a_t:", a_t)
print("b_t:", b_t)
```

This code defines a function update_state that takes in the state variables a and b, the control variables u and v, and the parameters wkv and torch.randn. It then uses the PyTorch torch.exp function to compute the exponential of the control variables and the log of the normalization factor. Finally, it uses these values to update the state.

**Explain the following metaphor: Life is like cats.**

The metaphor "Life is like cats" suggests that life can be unpredictable and full of unexpected twists and turns, just like a cat's behavior. It implies that we cannot fully control or predict the outcome of our lives, just as we cannot control the movements of a cat. The metaphor also implies that life is complex and multifaceted, with many different variables and factors at play. Just as a cat can have multiple personalities or moods, life can have many twists and turns that are difficult to predict or understand. Overall, the metaphor suggests that life is complex and unpredictable, much like a cat's behavior.

**RWKV-4-Raven-14B**

**Test the convergence of $\sum_{i=0}^{\infty} 2^{-i}$.**

The given series is a geometric series with the first term $a_0 = 2$ and common ratio $r = \frac{1}{2}$. We can find the sum of this series using the formula for a geometric series:

$$\sum_{i=0}^{\infty} a_i = \frac{a_0}{1-r} = \frac{\frac{1}{2}}{1-\frac{1}{2}} = \frac{1}{1-\frac{1}{2}} = \frac{2}{1} = 2$$

So, the sum of the given series is $\boxed{2}$.

---

**RWKV-4-Raven-14B**

**I would like to cook some Chinese food at home. Do you have any suggestions on the meal I could choose?**

Yes, of course! If you are not very experienced in cooking Chinese food, you could start with making noodles or rice, then make a dish using one of those. I also think making stir-fry with some meat is an excellent idea.

**I want to make something more special. What about red braised pork belly?**

Yes, red braised pork belly is also very popular and tasty.

**Tell me the steps to cook.**

You will need a few ingredients such as pork belly, star anise, sugar, ginger, scallions and wine. Put all of these in a pot, cook on medium heat and simmer until the meat is tender. After that, remove the meat from the pot and reserve the sauce. Slice the meat and add some ginger slices into it, then boil until it's almost done.

**What do I need to be careful with the most during the cooking process?**

To cook pork belly properly, it is very important to monitor the heat of the stove and be patient with the cooking process. Overcooking it can cause the pork to be tough, dry and difficult to chew. You can add a few ingredients like soy sauce or black beans to give the pork more flavor.

**Tell me more about the proper heat and timing.**

If the temperature of the water is too high, it can damage the collagen and connective tissues, resulting in tough pork belly. It is essential to control the temperature carefully, especially when braising the pork. Additionally, keep the water simmering but not boiling to maintain a low simmer, as high heat will dry out the pork belly and result in tough and unpalatable meat. Lastly, monitor the heat during cooking and make sure the heat is medium, as this is what is most appropriate for slow cooking pork belly.

**I see. Thank you for your advice.**

You're welcome! It was a pleasure chatting with you. Let me know if you need any further assistance in the future.

## RWKV-4-Raven-14B

**If all apples are fruits and all fruits are sweet, are all apples sweet? Please explain your answer in detail.**

Yes, all apples are fruits, and all fruits are sweet. Therefore, it is safe to assume that all apples are sweet.

**If birds can fly. Penguins are birds. Can penguins fly? Please explain your answer in detail.**

No, penguins cannot fly. Penguins are birds, but they are not capable of flight like other birds. They have wings and can glide for short distances, but they cannot take off or fly long distances like other birds.

**The day before two days after the day before tomorrow is Saturday. What day is it today? Please explain your answer in detail.**

Today is Friday. Two days after tomorrow is Sunday. Tomorrow is Saturday.

**If five cats can catch five mice in five minutes, how long will it take one cat to catch one mouse? Please explain your answer in detail.**

If five cats can catch five mice in five minutes, it means that each cat can catch one mouse in one minute. To find out how long it will take one cat to catch one mouse, we can use the concept of proportionality.
Let's assume that it takes x minutes for a cat to catch a mouse. Then, the number of mice caught by a cat in one minute is equal to the number of cats multiplied by the number of mice caught by each cat: x = 5 * 1 = 5
So, it will take 5 minutes for one cat to catch one mouse.

## RWKV-4-Raven-14B

**Summarize the following paragraph.**

**<The Abstract of this paper goes here>**

The paper presents a novel architecture, Receptance Weighted Key Value (RWKV), that combines the efficient parallelizable training of Transformers with the efficient inference of RNNs. The model is able to scale to tens of billions of parameters and exhibits linear computational complexity during training and inference, making it a promising alternative to Transformers for sequence processing tasks.

**Please further summarize it in one sentence.**

The researchers have developed a new architecture that combines the efficient parallelizable training of Transformers with the efficient inference of RNNs, which allows for linear scaling in memory and computational requirements.