# OpenReview forum: "RWKV: Reinventing RNNs for the Transformer Era"
_EMNLP/2023/Conference — EMNLP 2023 Findings_

### Official Review · Reviewer_HDNB · 2023-07-29

**Soundness:** 4

**Excitement:**

4: Strong: This paper deepens the understanding of some phenomenon or lowers the barriers to an existing research direction.

**Paper Topic And Main Contributions:**

- Transformers have significantly advanced various natural language processing (NLP) tasks. However, they suffer from high memory and computational complexity that grows quadratically with sequence length. On the other hand, recurrent neural networks (RNNs) exhibit linear scaling in memory and computational requirements but struggle to achieve the same performance as Transformers due to limitations in parallelization and scalability.

- To address these issues, the paper proposes a new model architecture called Receptance Weighted Key Value (RWKV). This model combines the efficient parallelizable training of Transformers with the efficient inference of RNNs. RWKV utilizes a linear attention mechanism, enabling us to formulate the model as either a Transformer or an RNN. This allows for parallel computations during training and maintains constant computational and memory complexity during inference.

- RWKV has successfully been scaled up to 14 billion parameters, making it the largest dense RNN ever trained. Remarkably, RWKV performs on par with similarly sized Transformers, indicating that this architecture could lead to more efficient models in the future. This work represents a significant step towards balancing the trade-offs between computational efficiency and model performance in sequence processing tasks.

**Questions For The Authors:**

- From the experiment results of MathQA, it seems RWKV performs poorly. Is those tasks need hard reasoning a intrinsic weakpoint of RWKV?
- How does the RWKV perform on long inputs?
- The paper mentions many techniques, are these universal across all sizes?

**Reasons To Accept:**

- This paper introduces RWKV, a groundbreaking architecture that effectively combines the strengths of Recurrent Neural Networks (RNNs) and Transformers while addressing their respective limitations.
- Detailed experiments demonstrates exceptional performance and efficiency on benchmark datasets for large-scale models, making it a promising solution for various natural language processing tasks.
- The authors release a set of pretrained models with varying parameter sizes, ranging from 169 million to 14 billion. This is useful for community research on high capacity modeling.

**Reasons To Reject:**

Recent large language models are using float16 or bfloat16 precision, it will be great to see RWKV also works in these precisons.

**Reproducibility:**

4: Could mostly reproduce the results, but there may be some variation because of sample variance or minor variations in their interpretation of the protocol or method.

**Reviewer Confidence:**

3: Pretty sure, but there's a chance I missed something. Although I have a good feel for this area in general, I did not carefully check the paper's details, e.g., the math, experimental design, or novelty.

---

> ### Author Rebuttal · Authors · 2023-08-29
>
> > Recent large language models are using float16 or bfloat16 precision, it will be great to see RWKV also works in these precisons.
>
> All RWKV models were trained using bf16. We will make this explicit in the paper.
>
> > From the experiment results of MathQA, it seems RWKV performs poorly. Is those tasks need hard reasoning a intrinsic weakpoint of RWKV?
>
> We briefly discuss the poor MathQA performance in Appendix I. In brief, we believe it may be caused by the order of information presented in the math questions of MathQA, and it’s unclear at this stage whether this issue is solvable by prompt engineering.
>
> > How does the RWKV perform on long inputs?
>
> We evaluate RWKV’s performance on long contexts in section 6 (see 6.2 in particular), and evaluate on the long-range arena benchmark in Appendix G.2
>
> > The paper mentions many techniques, are these universal across all sizes?
>
> Yes. All model architecture techniques listed in section 4 are model-size agnostic, and should benefit the model at any scale. The extent to which each technique benefits the loss at each scale is currently unknown, however, and an ablation study will be performed in future work to make this relationship explicit.

---

### Official Review · Reviewer_85wr · 2023-08-04

**Soundness:** 2

**Excitement:**

5: Transformative: This paper is likely to change its subfield or computational linguistics broadly. It should be considered for a best paper award. This paper changes the current understanding of some phenomenon, shows a widely held practice to be erroneous in someway, enables a promising direction of research for a (broad or narrow) topic, or creates an exciting new technique.

**Paper Topic And Main Contributions:**

This work presents RWKV an evolving model / open-source architecture that is meant to rival transformers while providing RNN-like compute. As the authors note this is the largest RNN model trained to date in NLP and shows strong and compelling performance. The paper contextualizes this model among the other models being studied and described in this space, and the historical models that people have proposed. Results show that there is compelling performance for the model that makes it worthy of continual study.

**Reasons To Accept:**

* This work should be accepted. The team here has tried something hard and ambitious, and succeeded in scaling this model to the point of good performance. The data points here are extremely valuable and I think this paper should be presented in an NLP venue. I have many issues with the form of the presentation, but I do not think my concerns should be seen as a vote for rejection.

* Section 4 is very nicely presented. Compared to the github for this project which is very challenging to understand, this section presents a coherent, readable and implementable description of the approach and its usage.


**Reasons To Reject:**

Given the above, I hope the authors take these concerns into account. There are many aspects of the work that I think are quite problematic, and without the scale and ambition of the other parts I would have not wanted it accepted. In general this paper reads like an advertisement and not scientific research. Captions and experimental detail is lacking throughout.  I hope they take the time to fix them.

* Figure 1 needs to have actual references to datasets and calculations. Having unlabeled graphs is not okay in a published paper. Languages need to be provided as well (BLOOM is multilingual, are these English tasks?)

* My understanding is that RWKV is roughly equivalent to the AFT local model that was previously presented. Yet this is not mentioned in the paper and the table does not include this key property. Is this an oversight or am I missing something?

* All the main information in the paper is shown in graphs in terms of scaling. While I understand why the authors want to show their model in this way, as a reader I want to see standard tables showing tokens / ppl (or bpc). Please include these tables in the paper so I can understand the data efficiency without trying to extrapolate from tables.

* While the pen-and-paper FLOP calculations are interesting, would be curious to understand how the actual training time compares on real hardware. Some graphs in the main paper would help.

* There are many different extensions to a raw RWKV architecture presented in this paper, but there are not any ablations in the body showing whether these were helpful or not in terms of loss.

* I do not understand the units or experimental setup of figure 6. Would love a transformer baseline as well to understand the curve.

Small:

* Figure 2 / 3 show the same thing roughly 3 times. At first read this was a bit confusing.

* How were the tasks chosen for the various analysis plots? Are these random or representative in some manner?

* Generally the graph labels are much too small to read, please increase these to be similar to the text itself.

* Can you provide more details on exactly the inference method / software hardware used for the text generation results? From the text it is unclear whether it is even cpu or gpu.

* How close do you think this model is to reaching the FLOP values presented in this work? Is this theoretical or is it comparable in a wall clock sense?


**Reproducibility:**

4: Could mostly reproduce the results, but there may be some variation because of sample variance or minor variations in their interpretation of the protocol or method.

**Reviewer Confidence:**

4: Quite sure. I tried to check the important points carefully. It's unlikely, though conceivable, that I missed something that should affect my ratings.

---

> ### Author Rebuttal · Authors · 2023-08-29
>
> We thank the reviewer for the detailed review! Please find below our response addressing the raised concerns:
>
> > Figure 1 needs to have actual references to datasets and calculations. Having unlabeled graphs is not okay in a published paper. Languages need to be provided as well (BLOOM is multilingual, are these English tasks?)
>
> We will add references for the final paper version. All languages are english unless explicitly stated otherwise.
>
> > My understanding is that RWKV is roughly equivalent to the AFT local model that was previously presented. Yet this is not mentioned in the paper and the table does not include this key property. Is this an oversight or am I missing something?
>
> AFT and RWKV are indeed overall similar, but differ in a few key ways. We compare these exact differences between architectures in section 4.1, but at a high level AFT learns a decay for each pair of locations, where in the local approach if the distance between two locations is higher than the kernel size this decay is 0. RWKV uses exponential decays that decreases with the distance. Other key differences are the relative positional information (which makes RWKV recurrent) and RWKV’s token-shift.
>
> > All the main information in the paper is shown in graphs in terms of scaling. While I understand why the authors want to show their model in this way, as a reader I want to see standard tables showing tokens / ppl (or bpc). Please include these tables in the paper so I can understand the data efficiency without trying to extrapolate from tables.
>
> We will include these tables in an appendix for the final paper version
>
> > While the pen-and-paper FLOP calculations are interesting, would be curious to understand how the actual training time compares on real hardware. Some graphs in the main paper would help.
>
> We agree that these results should be included. We will try to dig through experimental results and compare the actual training times, but if this can’t be reliably done (some results are not explicitly timestamped), we will have to leave this for future work.
>
> > There are many different extensions to a raw RWKV architecture presented in this paper, but there are not any ablations in the body showing whether these were helpful or not in terms of loss.
>
> We agree that ablations would be useful, and will attempt to collect the data necessary to add these figures for the final paper version. If we can’t collect the data in time, these results will be left for future work.
>
> > I do not understand the units or experimental setup of figure 6. Would love a transformer baseline as well to understand the curve.
>
>  We agree that Figure 6 appears unclear and we will improve it for the final paper version.
>
> > Small:
>
> These minor changes will be fixed for the final paper version. Thank you for tracking these down!

---

### Official Review · Reviewer_rSzx · 2023-08-07

**Typos Grammar Style And Presentation Improvements:** 1. L126
**Soundness:** 3

**Excitement:**

4: Strong: This paper deepens the understanding of some phenomenon or lowers the barriers to an existing research direction.

**Paper Topic And Main Contributions:**

This paper proposes an alternative architecture to transformers, inspired by RNNs, that is more memory- and compute-efficient. The main idea is to replace the quadratic complexity self-attention mechanism with a recurrent time mixing mechanism. While this time mixing component is not parallelizable over the sequence length dimension during training, it is parallelizable over the hidden dimension, enabling efficient training comparable to that of transformers. This work trains models of up to 14B parameters on the PILE dataset, showing comparable performance to transformers on various tasks. Experiments also confirm that the trained models exhibit linear time scaling concerning the number of tokens generated, as opposed to the quadratic scaling of existing transformer LMs.

**Reasons To Accept:**

1. The proposed architecture's comparable training speed to transformers, much faster inference, lower memory footprint, and potentially unlimited context window size make it an appealing alternative to transformers.
2. Empirical evidence demonstrates that the proposed method works well on large-scale models and training data.

**Reasons To Reject:**

1. The authors acknowledge an information bottleneck problem in the proposed architecture, as information from past context must flow through fixed-size hidden vectors. This issue might be reflected in the experiments in Appendix I, showing that the model is more sensitive to the prompt. This bottleneck could negatively impact tasks that require more fine-grained attention.
2. The time mixing component, while parallelizable along other dimensions, is not parallelizable in the time dimension. This lack of parallelization could become a training bottleneck for very long context windows.
3. Two specific tasks (ReCoRD and Winogrande, as shown in Figure 5) see the model underperforming other models. This underperformance requires further investigation.
4. A significant potential benefit of an RNN-like formulation is its applicability to longer contexts, but Figure 6 appears to limit this method to up to 2^12=4096 tokens. Further exploration of context length scaling is desirable. Additionally, most of Figure 6 is unsurprising, as more context naturally results in lower perplexity within the context window size. The figure's x-axis should start with the context window size being trained on. Clarification on the context size being fine-tuned up to would also be beneficial.

**Reproducibility:**

3: Could reproduce the results with some difficulty. The settings of parameters are underspecified or subjectively determined; the training/evaluation data are not widely available.

**Reviewer Confidence:**

4: Quite sure. I tried to check the important points carefully. It's unlikely, though conceivable, that I missed something that should affect my ratings.

---

> ### Author Rebuttal · Authors · 2023-08-29
>
> > The authors acknowledge an information bottleneck problem in the proposed architecture, as information from past context must flow through fixed-size hidden vectors. This issue might be reflected in the experiments in Appendix I, showing that the model is more sensitive to the prompt. This bottleneck could negatively impact tasks that require more fine-grained attention.
>
> We acknowledge this shortcoming, and hope to alleviate this issue in future iterations of RWKV.
>
> > The time mixing component, while parallelizable along other dimensions, is not parallelizable in the time dimension. This lack of parallelization could become a training bottleneck for very long context windows.
>
> Neither RWKV nor attention-based architectures GPT can improve in the time dimension beyond O(T) where T is the sequence length. Therefore, either both RWKV and GPT are parallelizable in the time dimension, or they both are not. We note that RWKV has a notable decrease in time and space complexity as T increases compared to competing architectures (see table 1), and this is a key strength of our approach.
>
> > Two specific tasks (ReCoRD and Winogrande, as shown in Figure 5) see the model underperforming other models. This underperformance requires further investigation.
>
> We agree with this shortcoming and are actively investigating.
>
> > A significant potential benefit of an RNN-like formulation is its applicability to longer contexts, but Figure 6 appears to limit this method to up to 2^12=4096 tokens. Further exploration of context length scaling is desirable. Additionally, most of Figure 6 is unsurprising, as more context naturally results in lower perplexity within the context window size. The figure's x-axis should start with the context window size being trained on. Clarification on the context size being fine-tuned up to would also be beneficial.
>
> We agree that exploring even longer contexts would be an interesting result, and we have trained models up to 128k context length. These results were taken after the anonymity period and will be announced more broadly after it ends.
> For previous attention-free models such as LSTMs, performance sometimes doesn’t improve beyond ~100 tokens. Our result verifies that RWKV doesn’t have this limitation.
> For the long-context finetuning experiments in section 6.2, we used the following context lengthening finetuning schedule for both RWKV 7B and 14B:
>
> 1024 -> 2048 for 10B tokens
> 2048 -> 4096 for 100B tokens
> 4096 -> 8192 for 100B tokens
>
> Then these finetuned models are evaluated on the Pile test set in Figure 6. We have explicitly noted this in Section 6.2 for the final paper version.
>
> > L126:a computationally efficient alternatives, L136:Simultaneously with this work, (Poli et al., 2023): citep -> citet
>
> Thanks for catching these! They have been fixed.

---

### Official Review · Reviewer_Zd3h · 2023-08-12

**Soundness:** 3

**Excitement:**

3: Ambivalent: It has merits (e.g., it reports state-of-the-art results, the idea is nice), but there are key weaknesses (e.g., it describes incremental work), and it can significantly benefit from another round of revision. However, I won't object to accepting it if my co-reviewers champion it.

**Paper Topic And Main Contributions:**

The paper discusses a novel model architecture called Receptance Weighted Key Value (RWKV) that combines the training efficiency of transformers with the inference efficiency of recurrent neural networks (RNNs). The approach utilizes a linear attention mechanism, allowing the model to be formulated as either a Transformer or an RNN. This enables parallel computations during training and maintains constant computational and memory complexity during inference. The authors conducted experiments with models containing up to 14 billion parameters and found that RWKV performs similarly to similarly sized Transformers. The main goal of this work is to find a balance between computational efficiency and model performance in sequence processing tasks.

**Questions For The Authors:**

If both long and short contexts require the same fixed amount of memory space, what implications could this have on the performance when dealing with long contexts?
How does RWKV compare to other non-Transformer architectures for sequence processing tasks?
How does RWKV scale to even larger models?
How does RWKV perform on tasks that require long-range dependencies, such as machine translation?


**Reasons To Accept:**

RWKV is good work because it is the initial step to fill the gap between RNN and Transformer and it is a novel model architecture that combines the efficient parallelizable training of Transformers with the efficient inference of RNNs. In addition, RWKV performs on par with similarly sized Transformers on a variety of natural language processing tasks, suggesting that it is a promising new architecture for sequence processing tasks.

**Reasons To Reject:**

It is not as flexible as Transformers. Transformers can be used for a wider variety of tasks, and they are more adaptable to different input and output formats. Transformers are able to learn more complex relationships between input and output sequences than RWKV. This is because Transformers' attention mechanism allows them to attend to arbitrary positions in the input sequence, while RWKV's linear attention mechanism is only able to attend to certain positions.

**Reproducibility:**

4: Could mostly reproduce the results, but there may be some variation because of sample variance or minor variations in their interpretation of the protocol or method.

**Reviewer Confidence:**

5: Positive that my evaluation is correct. I read the paper very carefully and I am very familiar with related work.

---

> ### Author Rebuttal · Authors · 2023-08-29
>
> We thank the reviewer for the thorough and careful review. Please find below our response addressing the raised concerns:
>
> > It is not as flexible as Transformers. Transformers can be used for a wider variety of tasks, and they are more adaptable to different input and output formats.
>
> >Transformers can be used for a wider variety of tasks, and they are more adaptable to different input and output formats. Transformers are able to learn more complex relationships between input and output sequences than RWKV. This is because Transformers' attention mechanism allows them to attend to arbitrary positions in the input sequence, while RWKV's linear attention mechanism is only able to attend to certain positions.
>
> Although we agree that the attention mechanism of transformers can indeed learn more complex relationships than RWKV in its current form, we believe our work is publishable for two reasons:
> We tackle one of the most-studied domains of transformers, language modeling, using an alternative architecture. This architecture is in line with other research results that showed that the attention matrix, learned by transformers in language modeling, is low-rank and could be approximated by simpler alternatives. Essentially, our work proposes RNNs as a good base architecture for language modeling and uses language modeling to demonstrate this.
> These are initial results, and we expect RWKV to improve over time. While the causality restriction is inherent to the RWKV architecture, we believe that RWKV can be extended to other tasks and input/output formats, and will explore this in future work.
>
> > If both long and short contexts require the same fixed amount of memory space, what implications could this have on the performance when dealing with long contexts? How does RWKV compare to other non-Transformer architectures for sequence processing tasks? How does RWKV scale to even larger models? How does RWKV perform on tasks that require long-range dependencies, such as machine translation?
>
> We discuss the performance of RWKV in its current form in the second paragraph of “Limitations”.
> We leave the comparison of RWKV to other non-transformer architectures to future work
> We scale RWKV across a wide range of model sizes, and use this to generate scaling laws that demonstrate RWKV follows the broad scaling trend of transformers (section 5.2). We did not train larger models, but can reasonably extrapolate their performance trend compared to trained models.
> We evaluated RWKV on the long range arena (LRA) benchmark in Appendix G.2, and found that it performs second only to the S4 model.
>
> Once again, we thank the reviewer for the thoughtful review and are open to answering any further questions or receiving additional feedback.

---

### Meta-Review · Area_Chair_KcYP · 2023-09-18

**Recommendation:** 3

**Metareview:**

Summary of the paper: RWKV is the largest RNN model trained to date in NLP that rivals transformers in performance. The results show that the model has impressive performance, making it a worthwhile subject of further study.

This manuscript has a lot of positives. The idea presented in the paper is very ambitious and relevant to the NLP community. The proposed method has comparable training speed as compared to transformers with much faster inference and lower memory footprint.

Some of the core criticisms of the paper are on the empirical evaluations and the paper not being well written. Multiple details are missing including experiments such as actual compute time comparison, evaluation beyond 4K token length to showcase the use of RNN style method. RNN style methods trade-off accuracy and compute time (because of information bottleneck), an evaluation of this trade-off would be an interesting addition.

---

### Decision · Program_Chairs · 2023-10-07

**Decision:**

Accept-Findings

**Comment:**

Summary of the paper: RWKV is the largest RNN model trained to date in NLP that rivals transformers in performance. The results show that the model has impressive performance, making it a worthwhile subject of further study.

This manuscript has a lot of positives. The idea presented in the paper is very ambitious and relevant to the NLP community. The proposed method has comparable training speed as compared to transformers with much faster inference and lower memory footprint.

Some of the core criticisms of the paper are on the empirical evaluations and the paper not being well written. Multiple details are missing including experiments such as actual compute time comparison, evaluation beyond 4K token length to showcase the use of RNN style method. RNN style methods trade-off accuracy and compute time (because of information bottleneck), an evaluation of this trade-off would be an interesting addition.